# Orchestrating performance of healthcare networks subjected to the compound events of natural disasters and pandemic

Emad M. Hassan[1] & Hussam N. Mahmoud [1✉]

The current COVID-19 pandemic has demonstrated the vulnerability of healthcare systems worldwide. When combined with natural disasters, pandemics can further strain an already exhausted healthcare system. To date, frameworks for quantifying the collective effect of the two events on hospitals are nonexistent. Moreover, analytical methods for capturing the dynamic spatiotemporal variability in capacity and demand of the healthcare system posed by different stressors are lacking. Here, we investigate the combined impact of wildfire and pandemic on a network of hospitals. We combine wildfire data with varying courses of the spread of COVID-19 to evaluate the effectiveness of different strategies for managing patient demand. We show that losing access to medical care is a function of the relative occurrence time between the two events and is substantial in some cases. By applying viable mitigation strategies and optimizing resource allocation, patient outcomes could be substantially improved under the combined hazards.

[1] Department of Civil and Environmental Engineering, Colorado State University, Fort Collins, CO, USA. ✉email: Hussam.Mahmoud@colostate.edu

The COVID-19 pandemic has resulted in unprecedented consequences on all aspects of life around the world. From fatalities to financial crises, the pandemic's socio-economic losses have been enormous and will continue to escalate. Managing risk and minimizing such losses has been the subject of various national and international debates and has proven challenging. The challenges are further amplified when considering the compound effect of pandemics and natural disasters[1,2], especially if the natural disaster has a relatively short return period such as hurricanes and wildfires[3]. The occurrence of both events simultaneously or spatially over time can have dire consequences on infrastructure, social, and economic institutions of communities, requiring prompt and timely prevention and control measures[4]. In particular, healthcare systems play a critical role in minimizing losses and saving as many lives as possible[5]. When under both events, managing healthcare systems could be challenging due to the various conflicting mitigation strategies required under each event individually. On the one hand, containing the epidemic might require a restriction of movement to prevent the disease's spread. On the other hand, a disaster could destroy homes and bring people together, causing them to violate such a requirement, which could intensify disease transmission. The coupling and interplay between these two phenomena can indeed be catastrophic. As such, devising effective policies to manage healthcare networks when they are bombarded by these compound hazards is ever pressing.

Damage to the built environment due to natural disasters can limit or halt the main services provided by critical and essential institutions, including healthcare systems. The impact on healthcare could be direct in the form of damage caused by the event or indirect due to damage to the supporting services and infrastructure[6–9]. Even in the absence of any damage, the healthcare system could be strained due to patient surge resulting from injuries[5,10]. Failure of the healthcare system to offer the expected medical services following extreme events can increase morbidity and mortality[5]. Extreme natural events impact healthcare systems differently. Wildfires, for example, can cause injuries, displace communities and reduce the air quality, which changes patients' distribution and increase the demand for healthcare facilities, especially due to the respiratory diagnosed patients, burns, and heat-induced illness[11,12]. The increase in frequency and intensity of wildfires due to climate change will continue to place substantial demand for healthcare services[3,13–15]. Even though wildfire ignitions are mainly due to human-related activities, the propagation of the fires in the wildland and inside communities is a natural phenomenon. Accordingly, wildfires are considered natural disasters, which has been recently acknowledged by the World Health Organization[16] and by the US Congress through the Wildfire Disaster Funding Act (H.R. 2862), which was introduced in 2019.

Unlike natural disasters, epidemics directly impact communities' well-being and bring the healthcare system to the forefront of required critical services[17]. Even though outbreaks do not directly impact the communities' built environment, they affect the services provided by different lifelines that support the healthcare system[18]. Amble evidence continues to highlight the challenges in managing healthcare systems due to the COVID-19 epidemic[19]. The virus has been spreading differently from one country to another, and the role of many parameters in accelerating or decelerating the rates of disease transmission is yet to be clear[20]. Among these parameters, population characteristics and effectiveness of Non-Pharmaceutical Interventions (NPI), including lockdown, quarantine, social distancing, wearing masks, and isolation of infected populations, are expected to influence the spread of the virus.

There has been a recent surge in research on the impact of natural disasters and epidemics, independently, on healthcare systems. For natural disasters, the studies focused on hospital damage and fragility assessment, surge and patient demand models, and estimation of recovery of functionality[8,21,22]. Many studies pertained to earthquakes[10,23,24], wildfire[12,25], and climate-related events[26,27] as hazards, focusing on a single hospital[7,8,21,22,28] and, in some cases, on a network of hospitals[10,29,30]. For pandemics, the researchers investigated the healthcare system challenges, the expected increase in patient demand, and the needed staffed beds[17,19,31]. These articles considered constant hospital capacity. In this study, however, the hospital capacity is defined as the maximum possible number of treated patients per day for each bed type throughout the pandemic. While previous studies provided ample opportunities for advancing health science to improve societies' well-being following extreme natural disasters and epidemics separately, no studies have yet addressed their collective impact on communities' healthcare systems. The importance of understating their collective impact on healthcare systems and the society overall is demonstrated by the unprecedented and long fire season that devastated many communities in the Midwest and the Western United States (US) in 2020 during the COVID-19 pandemic.

In this research, we investigate the readiness of a healthcare system in the face of wildfire during an epidemic. We use the 2018/2019 Camp Fire case scenario in Butte County, California, and we assume that the COVID-19 epidemic occurred around the same time frame. We modify an existing healthcare system model[10] to represent the acute care hospital network in Butte County and simulate its response to the wildfire and epidemic. The wildfire damage data are taken from published datasets. A modified Susceptible-Exposed-Infectious-Recovered (SEIR) disease transmission model is presented to determine the expected number of infected cases and classify them based on their hospitalization service need. The disease spread parameters are estimated from different countries and territories worldwide while considering the uncertainty associated with each age group's hospitalization rates. We then model how the wildfire affects the disease spread, healthcare network, and patients' distribution. Various patient demand categories are considered, including wildfire and epidemic related patients as well as regular patients. These demands are then dynamically distributed to the healthcare facilities using a patient-driven model. We conclude by discussing the effectiveness of different mitigation strategies in reducing the compound impact of wildfire and epidemic on the healthcare system.

## Results

**Model application.** Located in Northern California, Butte County is one of the most vulnerable communities to wildfires. The city of Paradise, in particular, has 95% of its population located in a very high wildfire-prone region[32]. In 2018, the County faced a catastrophic wildfire event called the Camp Fire, which destroyed most of Paradise[33] and forced many residents to either permanently or temporarily evacuate[34] to neighboring cities, Oroville and Chico. Not all evacuated individuals found housings or apartments, and many were forced to stay in shelters. The overcrowded shelters created an ideal place for disease transmission where for instance viruses such as Norovirus impacted the residents of at least one shelter in Chico[34].

California has many healthcare facilities distributed around the state, as shown in Fig. 1a. A higher concentration of such facilities can be seen in major counties, as displayed in Fig. 1b. In contrast, other counties have no licensed beds. However, smaller cities such as Paradise have a much lower number of healthcare facilities because of their limited population. Before the Camp Fire, only

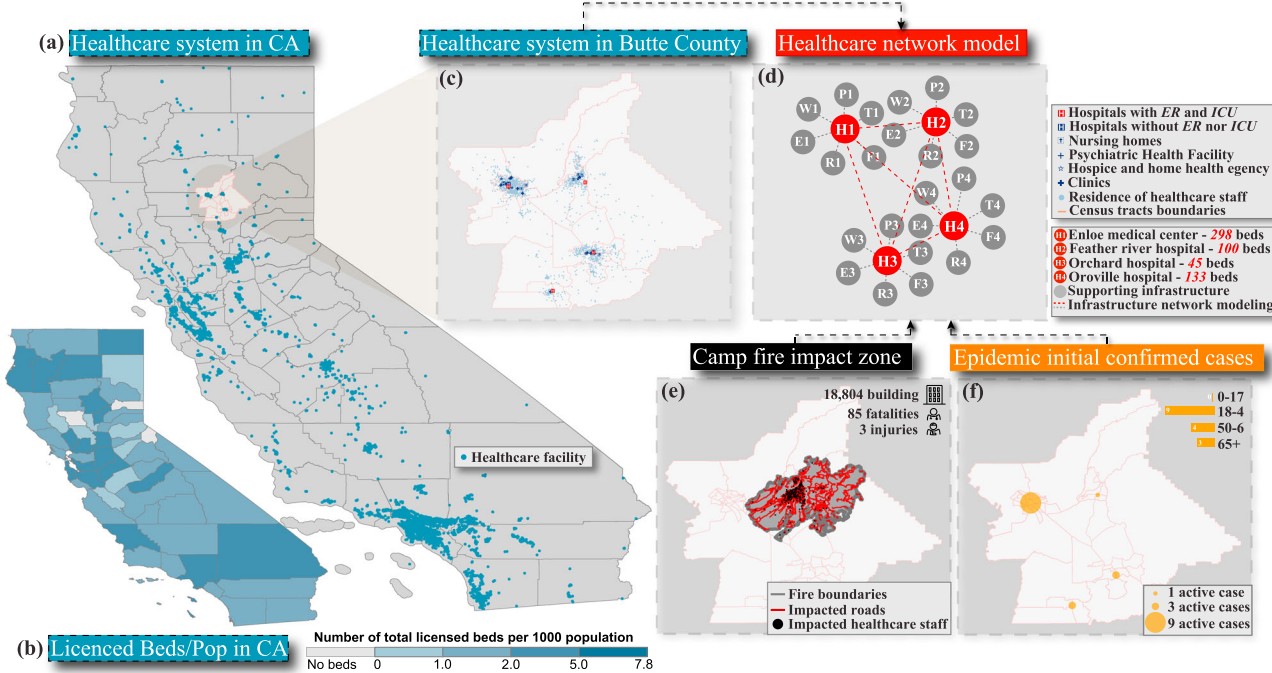

**Fig. 1 Healthcare network modeling in Butte County, CA, under the impact of the 2018 Camp Fire and the 2020 COVID-19 epidemic. a** Spatial distribution of healthcare facilities in California[40] and **b** the total number of licensed beds per 1000 population for California counties. **c** Location of different healthcare facilities in Butte County classified into those equipped with ER, and intensive care units, ICU; those without ER nor ICU; nursing homes; psychiatric health facilities; hospice and home health agencies; and clinics. Residence of the healthcare employees and staff according to OnTheMap website[38] for one year before the 2018 wildfire. **d** Depiction of the healthcare network, including the main healthcare providers and their initial capacities, staffed beds, and supporting infrastructure. The schematic shows the interaction between the healthcare facilities and between each facility and the supporting infrastructure[10]. **e** The Camp Fire impact zone, including the fire perimeter, impacted healthcare staff, and impacted transportation networks[33]. **f** Location and age distribution of the initial active cases for the COVID-19 epidemic in Butte County[39].

four healthcare facilities in the County were equipped with the units required to treat the critical case-patients, including emergency departments, ER, Intensive care units, ICU, and mechanical ventilators as noted in Fig. 1c, d. These included the Enloe Medical Center, H1, in Chico with 298 staffed beds[35], the Feather River Hospital, H2, in Paradise with 100 staffed beds, the Orchard Hospital, H3, in Gridley with 45 staffed beds[36], and the Oroville Hospital, H4, in Oroville with 133 staffed beds[37]. The Enloe Medical Center was the only hospital in Butte County equipped with a Level II trauma center and was provided with an air ambulance during the Camp Fire[35]. The residence addresses of each healthcare employee in Butte County Data before the wildfire are extracted from OnTheMap website[38], as shown in Fig. 1c. The network forming the healthcare system and the supporting infrastructure is depicted in Fig. 1d. The impacted regions in Butte County due to the 2018 wildfire and the 2020 COVID-19 are shown in Fig. 1e, f, respectively. The wildfire damaged many of the Feather River Hospital buildings[25], H2, and forced evacuation for the hospital patients and staff. All healthcare employees residing inside the Camp Fire perimeter were displaced, and all the roads located within the fire domain were closed, as shown in Fig. 1d. Because of the wildfire, the number of ER, inpatient, and ICU beds was immediately reduced, causing a significant drop in the healthcare system's functionality in Butte County. Initial cases of the COVID-19 outbreak are shown in Fig. 1f and were published by Butte County[39] on 7 May 2020 and are used as initial cases for the disease transmission model. For patient privacy, the county shared only the number and age distribution of the cases and not the exact location of these cases; therefore, the census tract for each case is assumed in this study.

**Compound impact of wildfire and pandemic**. Here, we discuss the individual as well as the collective impact of wildfire and COVID-19 epidemic on the healthcare system in Butte County. Following the 2018 Camp Fire, patient demand for hospitals increased due to the loss of 100 staffed beds because of damage sustained by the Feather River Hospital, evacuation of Paradise residents, degraded air quality, and overcrowding in shelters. The Camp Fire caused a total of 17 injuries that were mostly burns. Only two patients were admitted to the Enloe Medical Center and the other injuries received treatment at other burn centers in Northern California[41]. Other similar events have shown respiratory and asthma diagnoses to have increased by 34% and 112%, respectively, which raises the demand for the ER and inpatient beds during and for a short time after the wildfire[12]. Despite the effectiveness of the healthcare system in managing such demand, a substantial increase in ER visits of up to 12% was recorded at the Enloe Medical Center the year following the Camp Fire[42], which increased the average waiting time and decreased the patients' satisfaction[43,44]. In this study, our on average estimates, using the healthcare system network model outlined in Supplementary Fig. 1 and Supplementary Note 1, show the healthcare system functionality dropping by 18%, patients waiting time increasing by 35%, and patient treatment time reducing by 9% after the Camp Fire. The utilized patient-driven, which is a patient-centric health-seeking model, and the hospital interaction models show the inpatient's demand to increase by 14%, 13%, and 15% at the Enloe, the Orchard Hospital, and the Oroville Medical Center, respectively, compared with their demand before the wildfire (see Supplementary Fig. 2 and Supplementary Fig. 3). The continued closure of the Feather River Hospital after the Camp Fire resulted in a 17% reduction in

the total number of staffed beds in Butte County. Therefore, the impact of such closures on other healthcare facilities will be considered permanent for this study time frame (see Supplementary Fig. 4).

Suppose that the COVID-19 epidemic occurred in Butte County in 2018 instead of the Camp Fire with different disease spread scenarios representative of the U.S. and other impacted countries around the world (see Supplementary Fig. 5). Based on these scenarios and using the modified SEIR model, we calculate the number of different categories of COVID-19 hospitalization cases (see Supplementary Fig. 6). We find that the impact of the epidemic on the healthcare system is expected to increase the patient waiting time more than fivefold, reduce the patient treatment time to a minimum (i.e., medical staff spend less time treating patients to deal with the high demand), and result in many patients requiring ER to be sent home without treatments. Hospitals in Butte County will be overwhelmed for up to 135 days based on the disease transmission scenario used (see Supplementary Fig. 7).

Now, imagine the two events, the wildfire and the epidemic, take place in Butte County within a similar time frame. Assuming that the number of patients from the two events can be simply added or the healthcare system functionality can be determined by merely considering the two events separately could be misleading. Accordingly, the impact of wildfire and the COVID-19 epidemic on the healthcare system is captured simultaneously in this study. Each hospital is subcategorized into four different units: ER, Inpatient, ICU, and ICU with a mechanical ventilator. These units are expected to serve different patient categories, including regular as well as wildfire and pandemic-related patients. While regular patients are commonly distributed among the other units, the wildfire and COVID-19 related patients require respiratory treatment with more demand for mechanical ventilators and oxygen supplies. Here we assume that all evacuees are susceptible to disease transmission during the evacuation process, and those staying in shelters have a higher reproduction number compared with the remainder of the population who are practicing more restrictive measures. These circumstances will increase the total number of active cases and extend the disease peak period. To illustrate such interaction, we apply the US disease transmission rates ($\beta$, $\alpha$, $\gamma$, and $\delta$ see Methods) on Butte County's different census tracts and different group ages while assuming different wildfire occurrence time. We utilize the number of quarantined cases representing all COVID-19 patients who are infected and confirmed (tested positive) for comparison between different scenarios. Figure 2a shows a 35%, 59%, and 13% increase in the COVID-19 quarantined cases for the three different wildfire occurrence times, during and 15 days before and after the disease peak, respectively, compared with the epidemic only case.

The wildfire forced the Feather River Hospital evacuation and the residents of at least ten census tracts, which overcrowded other healthcare facilities within the county and increased disease spread of the Norovirus, especially in the ER[45]. In addition, the increase in asthma cases resulting from the excessive smoke during the wildfire increased the demand for hospital beds and limited the available beds for disease-related patients[12]. The analysis of the most probable hospital for each census tract (see Methods and Supplementary Note 1) after the wildfire highlights the expected change in demand for the healthcare facilities. That is, most residents in Paradise are using the Enloe Medical Center after the wildfire occurrence, as presented in Fig. 2b, c. It is also observed that the Oroville Hospital is the second choice for Paradise residents (see Supplementary Fig. 4).

The patients' distribution and effect on the healthcare system are sensitive to the occurrence time between the wildfire, $t_W$, and the epidemic, $t_E$. To illustrate such sensitivity, we show how the

distribution of the COVID-19 quarantined cases changes over time for the three wildfire occurrences in Fig. 2a. These patients are classified based on the hospitalization service needed and based on hospitalization rates from the Center for Disease Control (CDC)[46] and the European Center for Disease Prevention (ECDC)[47] into inpatient, ICU, and mechanical ventilator patients, as shown in Fig. 2d, e, and (f) for the three wildfire occurrence cases. These figures also display the envelope for the 2.5 and 97.5 percentiles for each hospitalization service based on hospitalization rates from ECDC[47] and Zhou et al.[48]. These pandemic-related patients are then distributed to different healthcare facilities and combined with their demand from regular and wildfire-related patients. The first case, wildfire @ Day 50, is when a wildfire occurs 15 days before the epidemic peak. In this case, the disease spread, especially within the evacuated census tracts and shelter residents, will be faster. This is because of the applied mitigation measures being insufficient at this stage to contain the disease spread and because the disease reproduction number is more than one coupled with infective cases being close to their peak. Consequently, a second wave of the disease spread occurs, which increases the average waiting time by about tenfold and the days in which the hospitals are overwhelmed to 245 as shown in Fig. 2g. The second case, wildfire @ Day 65, is when the wildfire takes place during the epidemic peak. In this scenario the disease spread will be limited due to the fact that strong mitigation measures, especially for the non-evacuees, are applied; therefore, the second wave of the disease spread is not as severe as the first case. The waiting time will increase to 8.3-fold and the hospitals will be overwhelmed for 223 days, as shown in Fig. 2h. Finally, the third case, wildfire @ Day 80, is when the wildfire occurs during the disease decline period. In such an event, while the number of evacuated patients is high, the disease reproduction number is less than one as a result of the strong NPI and the decrease in the active cases, reducing the number of patients without access to medical service in comparison to previous cases. However, the healthcare system will suffer during the epidemic peak, as shown in Fig. 2i, where the average waiting time will still be about five times higher than normal and the hospital will be overwhelmed for 189 days.

**Influence of events occurrence time**. Next, we examine the effect of different relative occurrence time between the wildfire and disease outbreak ($t_W - t_E$) coupled with different disease spread rates, obtained from six regions around the world—Hubei, China (S1), Iran (S2), Italy (S3), Spain (S4), Germany (S5), and the U.S. (S6), on the healthcare system. The ratios of untreated patients for each hospital services are used here to indicate the impact on the healthcare system. The ER and inpatient overflows are normalized using the maximum expected overflow for each disease spread scenario, as shown in Fig. 3a, b, respectively. In contrast, the ICU and mechanical ventilator maximum overflow are normalized by the healthcare system capacity, as shown in Fig. 3c, d, respectively. In this analysis, we assume that hospitals' effective disaster mitigation strategies implemented in previous events[23,49] are viable to use when combining wildfire and epidemic. These include, among other measures, the ability of (1) healthcare staff to reduce the treatment time, (2) hospitals to increase the number of physical beds for non-critical patients by using non-acute beds, unoccupied beds in other hospital units, non-fully featured ventilators as alternative beds, and (3) staff from the Feather River Hospital to temporarily close the gap in other healthcare staff shortage. The analysis shows that the total number of patients who will either leave without being seen or not have access to the ER is maximum for wildfire scenarios occurring during the epidemic acceleration phase, Fig. 3a. In this case, we consider the availability of backup beds for the ER departments and that all

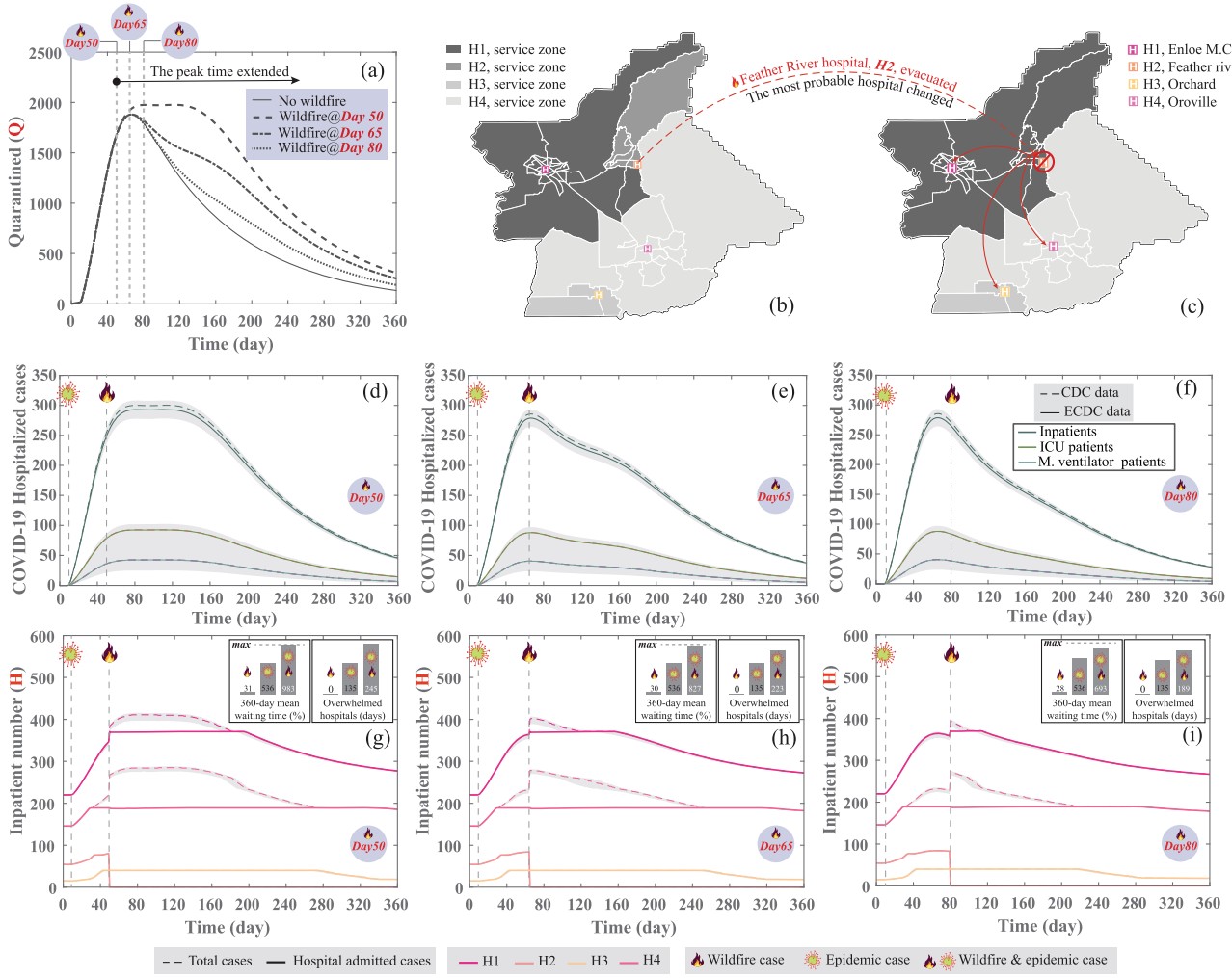

**Fig. 2 Compound effects of wildfire and pandemic. a** Impact of wildfire occurrence and the associated population evacuation on disease transmission in the community, which is highlighted using the distribution of the total quarantined cases that is assumed to be stricken by the wildfire before, during, and after the COVID-19 peak using the disease transmission rates of the U.S. The distribution of the most probable hospital per census tract for Butte County **b** before and **c** after the wildfire show that most patients in Paradise who have selected the Feather River Hospital before the fire are now choosing the Enloe Medical Center as an alternative. The distribution of COVID-19 related patients classified based on the hospitalization service needed for a different time, including **d** wildfire occurrence 40 days after the epidemic, **e** wildfire occurrence at the peak of the active epidemic cases, and **f** the wildfire occurrence during the disease decline period. The distribution of regular, wildfire and epidemic related patients on different hospitals in Butte County for the time after the epidemic using the disease transmission rates of the U.S. for a different time including **g** wildfire occurrence 40 days after the epidemic, **h** wildfire occurrence at the peak of the epidemic active cases, and **i** the wildfire occurrence during the disease decline period, 70 days after the outbreak. Note: time "10" marks the occurrence of the outbreak. These figures also show the 2.5 and 97.5 percentiles for hospitalization and inpatient cases, calculated using Monte-Carlo simulations with 100,000 trials.

alternative staff can handle the increase in demand, which enhances the capacity (number of treated patients per day) of the ER department to 600, 30, and 200 at H1, H3, and H4, respectively, boosting the maximum number of ER patients per bed per hour to 3. In the worst-case scenario, in a single day, the ER demand will be about double the capacity, which means thousands of patients will not have access to the ER services and will be sent home without treatment. For the inpatient beds, the hospitals are assumed to be able to enhance their operation by increasing room capacity, discharging non-critical patients early, and using available beds in other non-acute healthcare facilities. The analysis shows that the numbers of untreated patients increase when the wildfire occurs before the epidemic peak, and the inpatient demand will be 82% higher than the capacity in a single day, Fig. 3b. The availability of alternatives will be limited for the ICU beds and mechanical ventilators, especially if the disease also spreads outside the county; therefore, we only

consider using other hospital units and non-fully featured mechanical ventilators. The analysis highlights the mean and 95% confidence intervals of the maximum demand compared with the capacity for the patients who will need ICU beds, Fig. 3c, or will require a mechanical ventilator for different wildfire occurrence scenarios, Fig. 3d. The mean value of the ICU and mechanical ventilator demand overflow for the different disease spread scenarios developed from the six different regions mentioned above can overtop the capacity by 88% and 137%, respectively, in which the highest value of patient overflow will take place if the wildfire occurs 20 days after the disease outbreak.

## Discussion
Here we examine and discuss the effectiveness of other mitigation strategies, that if employed, might pay significant dividends in improving the functionality of the healthcare network and patients' accessibility to various medical services. These mitigation

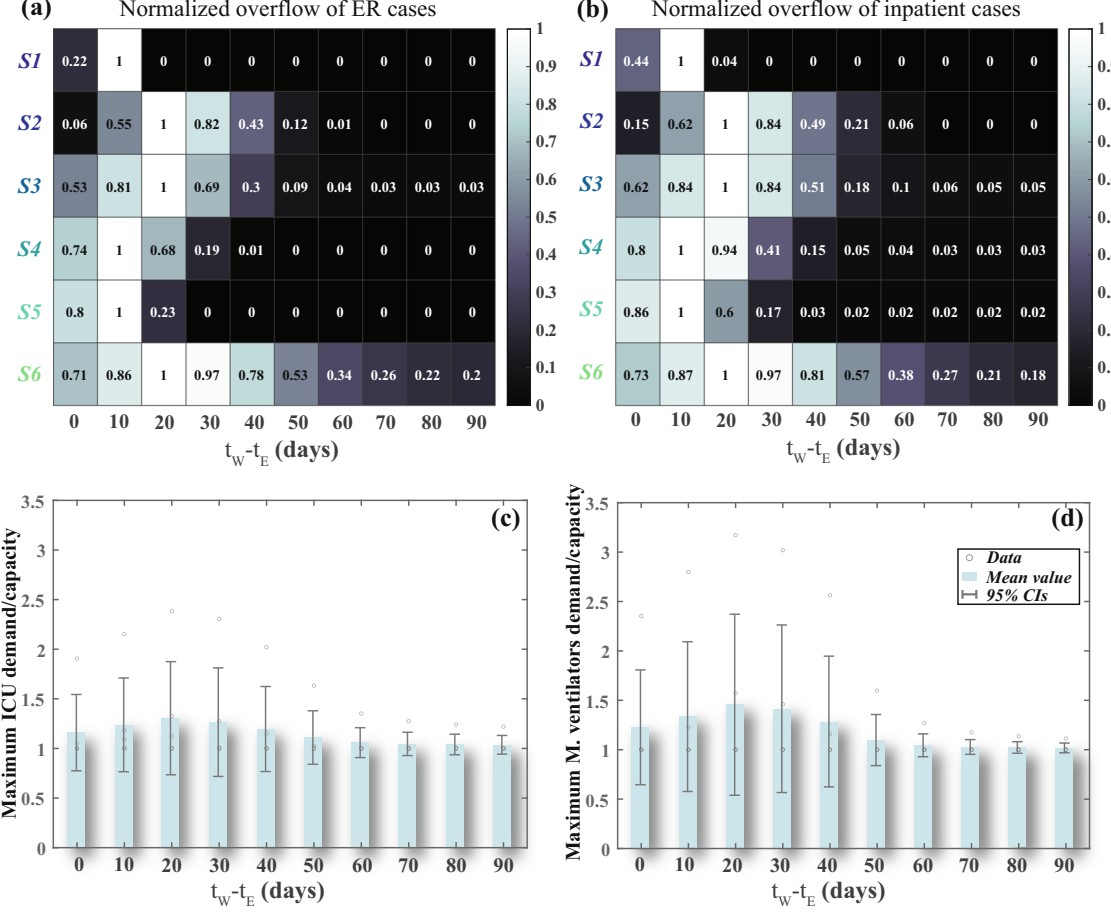

**Fig. 3 The effect of hazard relative occurrence time on hospital demand to capacity ratio. a** the normalized ratio of patients who leave the ER room without seeing a physician and **b** the normalized ratio of patients who require hospital admission and due to the higher demand for the healthcare system, they could not find a bed. The relationships between the ratio of maximum patient demand to the capacity and relative time between the wildfire occurrence, $t_W$, and disease outbreak, $t_E$, including **c** the patients who are supposed to stay in the ICU and **d** patients who require a mechanical ventilator. These relationships highlighted using the mean value and 95% confidence intervals (CIs) of the sample size ($n$) that represents six different disease spread rates obtained from Hubei, China (S1), Iran (S2), Italy (S3), Spain (S4), Germany (S5), and the U.S. (S6).

strategies focus on organizing the evacuation process, sheltering the wildfire evacuees, protecting the healthcare personnel, optimizing resource usage and allocation as well as increasing the number of temporary beds. To that end, we use the disease spread rates of the U.S. (S6), and we assume that the wildfire will occur 40 days after the disease outbreak (or 15 days before the disease peak), which we discussed in the "Results" section as a reference case to compare between different mitigation strategies. The wildfire evacuation process, especially for the Feather River Hospital, was chaotic in which not enough ambulances were available to transfer patients, and many of the designated evacuation roads were overcrowded[34]. Even though this chaos had a minor impact on patients, it could have been disastrous if it was coupled with the epidemic, see Fig. 2a. To reduce the disease spread during the evacuation process, we recommend using ambulances to evacuate the COVID-19 patients, redesign the evacuation roads to accommodate the evacuees, designate roads for emergency vehicles, and use protective measures for all the evacuees and emergency staff. By applying these strategies, we found that the patients' overflow for different services will reduce by more than 77% compared with the reference case due to the reduction in susceptible cases during the evacuation, which hinders disease spread during evacuation. This demand reduction will also lower the number of days in which hospitals are overwhelmed by 36%,

shorten the average waiting time by 35%, and increase the total functionality by 62%, see Fig. 4a.

As we discussed earlier, overcrowded shelters can substantially contribute to increasing the disease spread, see Fig. 2a. We suggest increasing the number of shelters to reduce the shelter occupant capacity and maintain at least a 6-ft social distance for all shelter residents, applying restrictive measures to the shelter residents, and enforcing isolation for all evacuees, and enhance testing and quarantine process. These strategies can reduce the patients' overflows for different services by more than 64% due to the decline in reproduction number among the shelter residents, and the deceleration of disease spread to the census tracts' residents where these shelters are located. Overall, the days in which hospitals are overwhelmed will reduce by 30%, the waiting time will decline by 28%, and the healthcare system's total functionality will increase by 50%, see Fig. 4b.

Providing appropriate and sufficient supplies to the frontline medical staff, especially protective equipment including respirators, gloves, face shields, gowns, and hand sanitizer[50], was critical in reducing their infection rates. We assume here that additional protection equipment can be used, which prevents disease spread in hospitals. To measure the impact of applying this strategy, we change the healthcare system staff's infection rates to be similar to that of other populations as opposed to the reported 19%

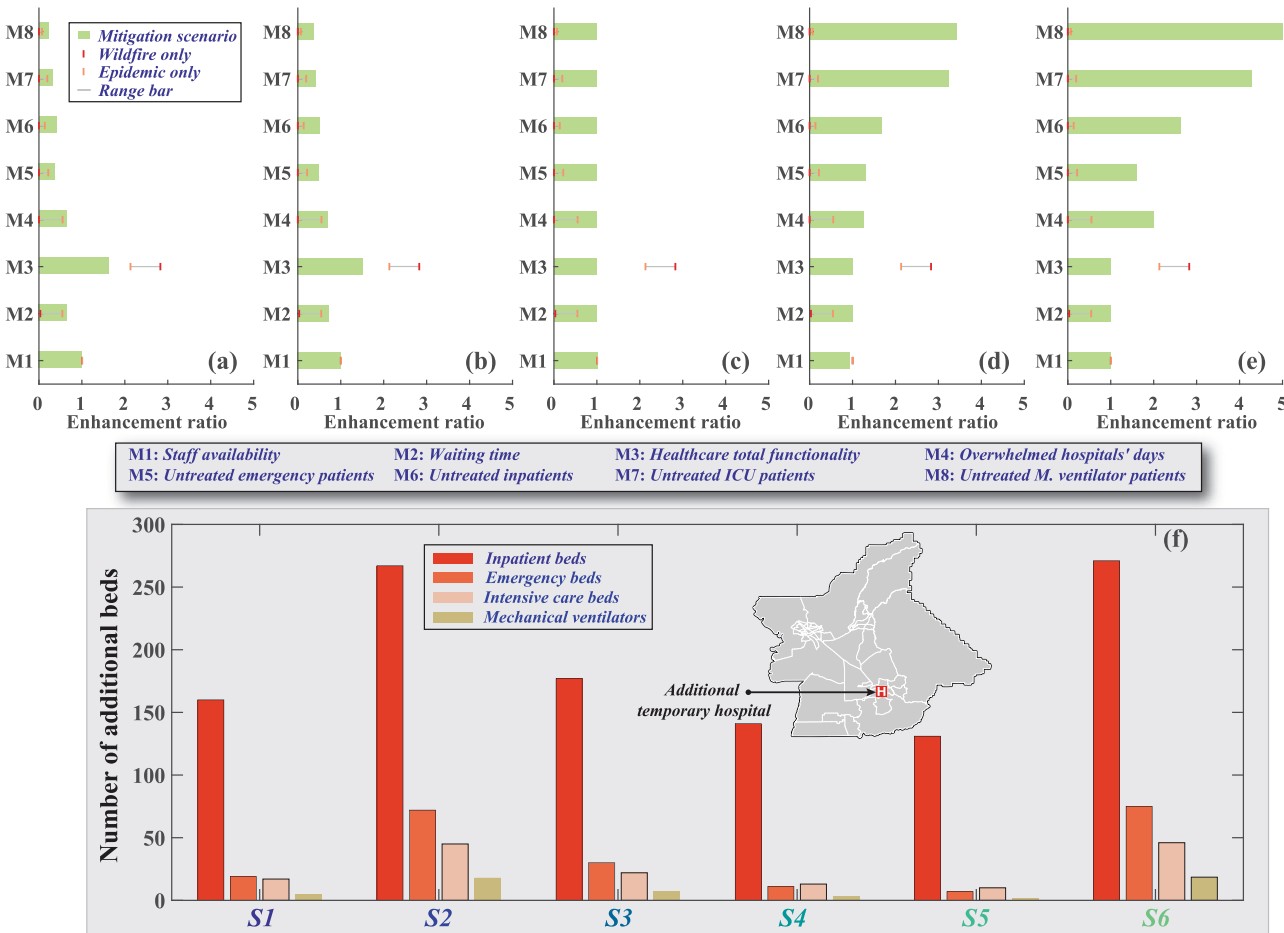

**Fig. 4 The effect of different mitigation strategies on healthcare system performance indicators.** These indicators including staff availability, waiting time, total functionality, and the total number of hospitals overwhelmed days and the patients' accessibility to medical services comprising ER, inpatient admission, ICU, and mechanical ventilators. The indicators are normalized by the reference case that used the US (S6) disease spread rates and assumed that the wildfire would occur 40 days after the disease outbreak. These normalized indicators are also compared with the normalized wildfire only and pandemic only scenarios and plotted as a range bar. These mitigation strategies are **a** applying stronger measures during the evacuation process, **b** provide more protections for the shelter residents, and **c** provide more protection for the frontline medical staff. We also show the effectiveness of other mitigation strategies we implemented in the analysis, including **d** replacing staff and **e** using non-acute care beds for non-critical inpatient cases. **f** We also present the optimization results for the location and number of different staffed beds needed for each of the considered disease spread scenarios.

infection rate for healthcare personnel[51]. This strategy will slightly reduce staff shortage by 0.12%, which reduces the waiting time by 0.1% and reduces the untreated patients by up to 1.2% by increasing the number of available beds, see Fig. 4c. These enhancements are minute, indicating the sufficiency of using alternative staff in addition to transferring staff from the Feather River Hospital in reducing the impact of staff shortage as well as the effectiveness of the current personal protection in the U.S.

In presenting the results, we assumed that the healthcare system is efficiently managed where hospitals have well-established approaches to address any shortages in staff, space, or supplies. However, this is not always the case. One of the main strategies that we assumed is that staff transfer from the Feather River Hospital to other facilities to substitute those who lost their homes in the wildfire or became infected after the wildfire or epidemic will be smooth and immediate. To quantify the effectiveness of this strategy, we assume no staff replacement will take place. The analysis shows that the average staff shortage will be 6%, which increases the hospitals' overwhelmed days by 27%, increases the waiting time by 3%, and reduces the total functionality by 26%, as shown in Fig. 4d. If the hospitals do not cover the ICU department's staff shortage, the number of untreated

ICU patients in this department will increase by 3.25-fold. In addition, we previously assumed that non-acute care beds could be used for non-critical patients as one of the mitigation strategies. To highlight this strategy's impact and the impact of not considering these beds on patients and the healthcare system, we assume no additional beds will be added. The analysis shows an increase in the number of untreated patients, which resulted directly from the absence of these beds. We also note that the days in which hospitals are overwhelmed increase by 100%, see Fig. 4e.

Understanding and identifying the components of the healthcare system that might be needed in the future is essential to mitigate the reduction in functionality and the number of untreated patients. In some cases, available community resources can be sufficient to accommodate the expected number of patients; however, in other cases, additional support, including staffed beds, mechanical ventilators, and supplies might be required. To investigate the quantity of the additional resources needed for each scenario, we assume that a temporary backup hospital can be added to the healthcare system. We use linear optimization to measure the optimal size and location of this hospital with two objective functions: (1) minimize the number of

untreated patients and (2) reduce the average waiting time for patients, while only considering the mitigation strategies used in the results sections and shown in Fig. 4e. We also impose a constraint in the analysis where non-acute care beds and unoccupied beds in other hospital units and non-fully featured ventilators will not be used by hospitals to close the gap in beds or mechanical ventilators shortage. The optimal location for the temporary hospital is in the city of Oroville, and the number of staffed beds and mechanical ventilators required range from 148 (using Germany data) to 392 (using U.S. data) beds, including ER, inpatient, and ICU beds and up to 19 ventilators. We want to clarify that the ratio between the bed types in Fig. 4f, is not typical for conventional hospitals. However, such ratio can still be achieved using a temporary field hospital that lasts during each investigated scenario's peak and can be reduced if the patient demand is lowered.

In conclusion, we devised a new framework for investigating the compound impact of wildfire and epidemics on a healthcare system. To highlight the viability of the proposed approach, we used Butte County, California as a testbed community and coupled the 2018 Camp Fire that occurred in Paradise with the current 2020 COVID-19 pandemic. To assess the sensitivity of healthcare performance to different scenarios, we investigated different disease transmission cases and various relative occurrence time between the two events. We showed that each event individually could indeed strain the healthcare system; however, combining the two events can leave an unprecedented and enormous impact on patients and the healthcare system. We then explored the effect of different mitigation approaches on reducing such impact and found that applying restrictive measures, especially during the evacuation process, and protecting the shelter residents as well as expanding the healthcare capacity by using non-acute beds can be viable mitigation strategies for substantially improving patient outcomes under the combined events of wildfires and epidemics.

This study focused on how a healthcare system might be impacted when a wildfire, as an example of a natural disaster, strikes a community that is suffering from an epidemic. While the results obtained highlight the significant importance of such an issue, we acknowledge that different communities can behave contrarily based on the type of natural disaster faced, epidemic characteristics, and the fabric of their social and economic institutions. Various protective measures and the individuals' adherence to these measures, which vary depending on the socio-cultural characteristics of communities, can have a different impact on disease spread[52]. Furthermore, we assumed that the evacuees would stay in Butte County with no exposures from migrants entering the County due to lockdown, and no vaccine will be available during the study time frame. These two assumptions will greatly impact the spread of disease in the community and might change the results and overall conclusions. The wildfire data and the disease transmission model parameters were derived from published data and reports that are limited to the study time. Using more data could lower the level of uncertainties associated with disease transmission estimates.

## Methods

**Healthcare system model for Butte County, California**. The functionality of healthcare facilities can be defined as the proportion of services offered by the facility and can be measured using different quantitative and qualitative performance indicators[7]. The service quantity refers to the number of patients treated while the quality represents the patient's satisfaction with the offered medical service. During major events such as wildfires and pandemics, hospitals' medical services such as emergency, inpatient, intensive care, and ventilators are critical. Therefore, we focused in this study on hospitals as a premier healthcare facility and the main provider for these services. Other healthcare facilities play an important role in lighting the burden on hospitals by accepting moderate patients.

Accordingly, in this study, we included other non-acute healthcare facilities associated with the hospitals. Inclusion of all healthcare facilities in the community, however, was not considered. To determine the functionality of a healthcare facility, we rely on a model recently developed by Hassan and Mahmoud[10], where quantity and quality measures of the offered medical services by each facility are integrated (see Supplementary Note 1). The quantity is described by the number of staffed beds, which is a function of the staff, space, and supplies available at each hospital while considering the functionality of the supporting infrastructure such as water, power, transportation, and medical and non-medical supplies. On the other hand, quality is described by the patients' outcomes, including accessibility and effectiveness of the medical service, which is measured using patient waiting and treatment time. The patient waiting time refers to the time a patient remains before seen by medical staff and is described by the basic waiting time, travel time, and delay resulting from loss of beds or an increase in patient demand. The treatment time represents the time required to achieve patient outcomes and is affected by the available beds and physicians, hospital demand, and patient case criticality. The model[10] considers alternatives for the staff, supporting infrastructure, supplies, and the possibility of finding alternative beds for non-critical cases and is structured based on recommendations and guidelines[53–55], lessons learned from past events[23,56–58], and previously introduced analytical and theoretical models[9,22,29,59]. Logical relationships, i.e., success trees, between the model components, are utilized[7,10]. The main model parameters are summarized in Supplementary Table 1. The model results are verified with the data collected during the Camp Fire event[35–37,42,60]. Further refinement of the healthcare model should be explored in the future to include a comprehensive supply-chain network model, detailed spatial simulations of different hospital units, and considerations for the patient-internal delay time.

Different modifications are made to the Hassan and Mahmoud model[10] to allow for assessment of the impact of wildfire and epidemic on hospitals. The changes include using different success trees for each hospital service considered in this study (ER department and inpatient beds as well as ICU beds without mechanical ventilators and with mechanical ventilators). Even though these success trees have the same structure, the main events are calculated differently, and weighting factors are assigned to these main events to model the different needs for staff, space, and supplies for different bed types (see Supplementary Note 1). The data used in the functionality models are collected from the community health need assessment reports[4–6], determined using travel time estimation analysis[61], and obtained from online sources[43,44].

The hospital demand in the absence of stressors is classified based on the required medical service into ER visits, inpatient admission, and ICU admission. Some patients in the ICUs will also require mechanical ventilators, which is assumed, based on the national average, as 29% of the ICU patients[62]. The number of ER visits and inpatient admission are collected from the California Department of Health[60], while the data for the acute care admission are obtained from the California Health and Human Services Open Data Portal (OSHPD)[42] for the first half of the year 2018. We use these data to validate the patient-driven model (see Supplementary Note 2) utilized in this study to obtain the demand for each hospital in Butte county, which is then compared with the model developed by Jia et al.[63] (see Supplementary Fig. 8). The patient-driven model[10] shows that the inpatients' admission in Butte County is distributed as 0.51, 0.13, 0.01, and 0.35 for the Enloe Medical Center, the Feather River Hospital, the Orchard Hospital, and the Oroville Hospital, respectively, which are the same ratios reported before the Camp Fire occurrence. We used actual data for healthcare providers[35–37,44], medical insurance[35–37], transportation network and travel time[61], and population demographics[64] in Butte County before the Camp Fire as an input for this patient-driven model.

The complex network interaction between hospitals, including patients, staff, supplies, and resource transfer, is considered in the utilized model. The patient transfer can redistribute demand for healthcare facilities, reduce patient waiting time in the ER, and provide alternatives for patients admitted to overwhelmed hospitals. The probability of patient transfer is subjected to patient constraints, receiver hospital constraints, the availability of transportation methods especially for patients with critical cases, and a functional telecommunication network to transfer the patient records (see Supplementary Fig. 3). The utilized model also considers staff and resources transfer (see Supplementary Note 3).

**Wildfire hazard impact model**. For 17 days, the Camp Fire was active in the city of Paradise, Butte County, CA, and has been recognized as the most destructive fire in California history[34]. The fire resulted in 150,000 acres burned, over 18,000 buildings destroyed, 85 persons killed, and the whole Paradise community (about 40,000 residents) displaced. The wildfire not only damaged the built environment, but also social and economic institutions were severely impacted. Most of Paradise residents were dislocated to nearby cities, and some stayed in shelters located in Oroville and Chico. Our focus in the wildfire model is on its impact on the healthcare system.

The Camp Fire damaged part of the Feather River Hospital campus, and the hospital was evacuated on 8 November 2018. This hospital was one of the four acute care hospitals in Butte County. It contained 17% of the county's staffed beds and offered medical services for about 13% of the county's patients[42]. After the evacuation, an increase in the number of inpatients in nearby hospitals was

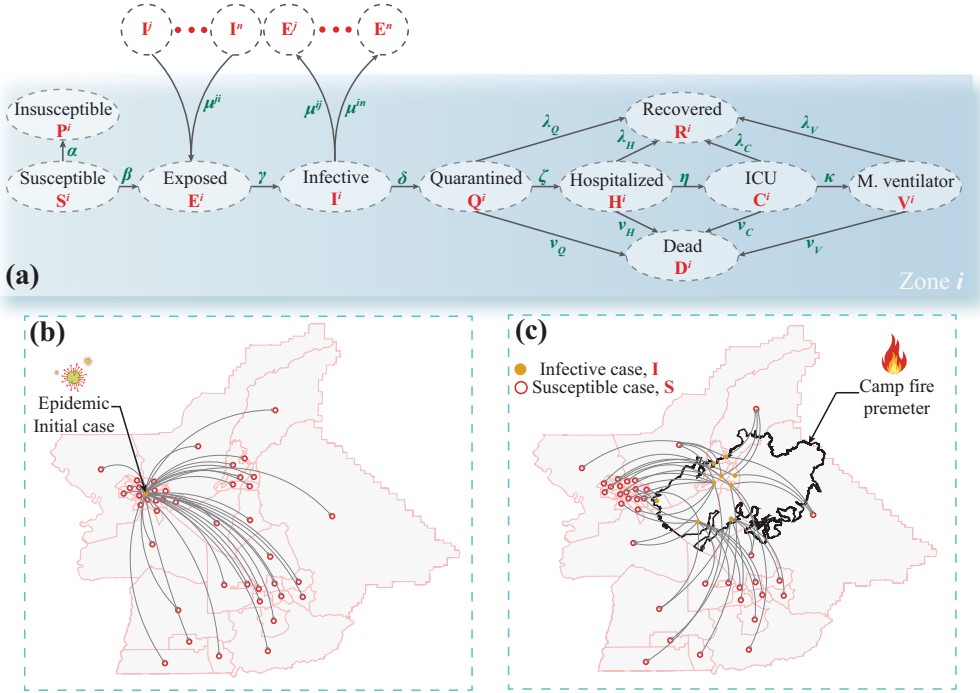

**Fig. 5 Modified SEIR model. a** The disease transmission model extending the classical SEIR model where the population in a census tract zone *i* is categorized into ten different cases while accounting for the possibility of disease spread (**b**) from infected to non-infected census tracts as well as **c** between the population from the evacuated zones, after the natural disaster, and their host zones.

recorded, especially at the Enloe Medical Center[42]. The published data set of building status after the Camp Fire[33] are used with the residence addresses of the healthcare employees and staff[38] to estimates the number of healthcare personnel who were displaced after the wildfire, which reduced the staff availability in the other healthcare facilities and impacted the overall hospital functionality (see Supplementary Note 1). During the wildfire, the transportation network within the Camp Fire was closed, which impacted patients' accessibility to medical service and increased the patient travel time (see Supplementary Note 1). We also considered the power outage and natural gas shortage as well as the disturbance in medical and non-medical supplies during the Camp Fire and their impact on hospital functionality and the total number of available staffed beds (see Supplementary Note 1). The wildfire can also increase ER visits and inpatient admission[12] as a result of the surge in patients with burns and respiratory symptoms[11]. The number of burned patients is collected from published data[41], while the number of wildfire smoke-related patients during and after the Camp Fire are estimated, due to the limited data available, based on the mean percentage of increase in ER visits and inpatient admission published for the San Diego 2007 wildfire[12].

**Disease transmission model formulation.** Different disease transmission models have been used in previous studies to simulate the impact of various epidemics on population[65]. The SEIR model is widely used to simulate the spread of COVID-19[66,67]. Other researchers introduced special disease transmission models[68,69]. The disease transmission model used in this study, Fig. 5a, is based on the classical SEIR model, which was modified to include ten different states as follow: susceptible, **S**, insusceptible, **P**, exposed, **E**, infective, **I**, quarantined, **Q**, hospitalized, **H**, ICU admitted, **C**, mechanical ventilator, **V**, recovered, **R**, and deceased cases, **D**. All ten cases are a function of time, *t*, after observing the initial confirmed cases, we call it in this study disease outbreak. The infected cases are first categorized into infective, self-quarantined, and hospital admitted. These categories simulate the effect of quarantine on disease spread and are used to distinguish between different sub-categories of confirmed cases (**Q**, **H**, **C**, and **V**) based on their medical need. In which, **Q** is for cases with mild or no symptoms, **H** is for cases needing hospital admission, **C** is for cases needing ICU, and **V** is for cases needing mechanical ventilators. These four states can be aggregated to represent the total number of active cases, **A**, that represents all the positive (confirmed) cases with no outcomes (recovered or deceased) yet. The additional cases (**Q**, **H**, **C**, and **V**) are used to subcategorize the original case **Q** in the classical SEIR model. They are utilized to determine the number of active epidemic cases that require different hospitalization services. The following model is constructed for population, *N*, of each census tract zone, *i*, in the investigated county with a total number of census tracts, *M*, while accounting for individuals visiting from the surrounding census tracts, *j*, who temporally reside in zone *i* and are in the infective state as shown in Fig. 5b, in which *j* ∈ [1:M] except *i*. The differential equations below are used to determine the

total number in each case at the census tract *i*.

$$dS/dt = -\beta(SI)/N - \beta S/N \sum \mu_{j \to i} I_j - \alpha S, j \in [1:M] - \{i\} \quad (1)$$

$$dP/dt = \alpha S \quad (2)$$

$$dE/dt = \beta(SI)/N + \beta S/N \sum \mu_{j \to i} I_j - \gamma E, j \in [1:M] - \{i\} \quad (3)$$

$$dI/dt = \gamma E - \delta I \quad (4)$$

$$dQ/dt = \delta I - \zeta Q - \lambda_Q Q - \nu_Q Q \quad (5)$$

$$dH/dt = \zeta Q - \eta H - \lambda_H H - \nu_H H \quad (6)$$

$$dC/dt = \eta H - \kappa C - \lambda_C C - \nu_C C \quad (7)$$

$$dV/dt = \kappa C - \lambda_V V - \nu_V V \quad (8)$$

$$dR/dt = \lambda_Q Q + \lambda_H H + \lambda_C C + \lambda_V V \quad (9)$$

$$dD/dt = \nu_Q Q + \nu_H H + \nu_C C + \nu_V V \quad (10)$$

where, $\beta$ is the infection rate, $\mu_{j \to i}$ is the rate of travel from zone *j* to the investigated zone *i*, $\alpha$ is the protection rate, $1/\gamma$ is the average incubation period, $1/\delta$ is the average quarantine time, $\zeta$ is the hospitalization rate, $\eta$ is the ICU rate, and $\kappa$ the mechanical ventilator rate. In addition, $\lambda_Q$, $\lambda_H$, $\lambda_C$, and $\lambda_V$ are the recovery rate for quarantined, hospitalized, ICU, and mechanical ventilator cases, respectively. Moreover, $\nu_Q$, $\nu_H$, $\nu_C$, and $\nu_V$ are the death rate for the quarantined, hospitalized, and patients in the ICU, and those on mechanical ventilators, respectively. The parameters $\zeta$, $\eta$, $\kappa$, $\lambda(s)$, and $\nu(s)$ are a function of the case age. The rate of travel $\mu_{j \to i}$ is assumed based on travel time between the census tract in which higher rates are assigned for the nearby census tracts. Furthermore, the basic reproduction number, $R_0$, is the average number of secondary infective cases produced by one infective case in the same census tract during the infectious period of this case and equals to $\beta/\delta(1-\alpha)^t$.

The positive values that are assigned to protection rate $\alpha$ are used to mimic community measures including lockdown, social distancing, wearing protective masks, among others, and also to make the basic reproduction number, $R_0$, a function of time. These mitigation strategies are expected to increase with time[70] and can successfully reduce the number of susceptible cases and the reproduction number likewise[71]. However, to simulate the disturbance and chaos associated with the evacuation process during the wildfire on the disease transmission, as shown in Fig. 5c, we model the population in the evacuated census tracts during the evacuation as a non-protected population (all became susceptible). In addition, as a

consequence of the low protection in shelters[34] that hosted most of the evacuees, the protection rates for the shelter residents are assumed to be less than others; therefore, we reset the basic reproduction number for these shelter residents to the initial time. The utilized model assumes a constant population for each investigated census tract over the epidemic time, $N$, which satisfies the equilibrium of $N = S + P + E + I + Q + H + C + V + R + D$ at any time $t$.

**Epidemic in Butte County.** COVID-19 has shown different transmission trends in different locations around the world, which increases uncertainties in forecasting models. To overcome this problem and to consider different disease transmission possibilities, six different scenarios are utilized in this study, including Hubei in China, Iran, Italy, Spain, Germany, and the U.S. Curve fitting for published active, recovered, and death numbers[72,73] are used to estimate the disease transmission parameters including $\beta$, $\alpha$, $\gamma$, and $\delta$ as shown in Supplementary Fig. 5. The initial incubation period is assumed based on the median estimated by Lauer et al.[74]. In addition to the initial quarantined cases announced by the County Fig. 1e and the population demographics for the year 2018[64], these parameters are used to simulate disease transmission in Butte County and estimate the number of quarantined cases in each census tract and each age group. The main disease transmission parameters, including $\zeta$ and $\eta$ and the patient length of stay, are listed in Supplementary Table 2. We also utilized normal distributions to simulate $\zeta$ and $\eta$ for each age group based on the data collected from ECDC[47] as shown in Supplementary Table 3. For $\kappa$, the beta distribution is used with 0.46, 5.22, and 3.08 for base, shape, and scale parameters, respectively[17,48]. The statistical distributions are employed to model the uncertainty associated with the number of different hospitalization cases using Monte-Carlo simulations with 100,000 trials. The parameters $\lambda(s)$ and $\nu(s)$ are estimated based on the data published by the CDC[46,51] for recovery and mortality rates per age group and other published research articles[48,74] for the rates of ICU patients and patients needing mechanical ventilators. These parameters are utilized to subcategorize the number of quarantined patients into cases visiting the ER, cases entering as hospital inpatients, cases requiring ICU, other cases needing ventilators, and the number of cases deemed closed as recovered or deceased. The average length of stay for COVID-19 patients at a hospital as an inpatient, ICU, and on a mechanical ventilator is assumed based on Zhou et al.[48] and Weissman et al.[17], in which the utilized median hospital length of stay is 12 days and the median duration of ICU stay is eight days while 75% of ICU stay will be on the mechanical ventilator as shown in Supplementary Table 2. Whereas for regular and wildfire-related patients, the data published by OSHPD[42] are utilized, as shown in Supplementary Table 1. These lengths of stay are then modified to account for the disturbance that might occurs during transferring patients from one hospital unit to another, which takes place when the subsequent unit is full. The healthcare staff has more exposure than others[51]; therefore, the percentage of healthcare workers infected in each census tract, and each age group is assumed based on the CDC report[51].

The total number of patients per census tract includes (a) the daily regular demand, (b) the wildfire smoke-related demand, and (c) the demand associated with the epidemic. This demand is used as input for the patient-driven model (see Supplementary Note 2), which is used to distribute the patients to the healthcare facility based on their functionality (see Supplementary Note 1). When the healthcare facility cannot treat patients, these patients can be transferred to another facility (see Supplementary Note 3). In the case where no beds are available for the patients, we consider them as untreated patients.

**Reporting summary.** Further information on research design is available in the Nature Research Reporting Summary linked to this article.

## Data availability
The base maps for California and Butte County are open-source maps provided by U.S. Census Bureau[75] while population demographics are available from the U.S. Census Bureau American Community Survey[64]. The location of the healthcare facilities in California is provided as an open-source shapefile by the Office of Statewide Health Planning and Development[40]. The number of active, recovered, and death COVID-19 cases for the six utilized disease spread scenarios were downloaded from real-time and dashboards sources[72,73]. Butte County related data are all available including transportation network data[61], healthcare employees data[38], Campfire incident data[33], and initial infected cases[39]. These data were used in the analysis and are included in the paper and the Supplementary Information. The data required for all simulations can be found electronically at https:// github.com/EmadShafik/Healthcare_NaturalDisaster_Pandemic/tree/v1.0.0 with the following DOI: 10.5281/zenodo.4417732.

## Code availability
The model components have been described in the paper and the Supplementary Information. The code required for analysis can be found at 10.5281/zenodo.4417732.

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

## Acknowledgements

Funding for this study was in part provided by the cooperative agreement 70NANB15H044 between the National Institute of Standards and Technology (NIST) and Colorado State University. The content expressed in this paper are the views of the authors and do not necessarily represent the opinions or views of NIST or the US Department of Commerce.

## Author contributions

E.M.H. and H.N.M. conceived the idea and contributed to the final version of the manuscript. E.M.H. carried out the simulations. H.N.M. supervised the project.

## Competing interests

The authors declare no competing interests.
