## [Peer Review File · Nature Communications]

REVIEWER COMMENTS

Reviewer #1 (Remarks to the Author):

Thank you for this interesting submission. Glad to see more work is being conducted on the resilience of healthcare to disasters. The manuscript presents the findings of a set of simulations about the impact of fire and the pandemic COVID-19 on the resilience of healthcare systems. Although this an interesting point of view, there are some major and fundamental issues that have to be addressed before any recommendations for publication is given.

The manuscript has many typos or maybe indications of lack of understanding of what the research is about. For example, in the abstract we find "Here, we investigate the combined impact of natural disasters and pandemic on the preparedness of a network of hospitals". I am not sure how you can investigate the impact of disasters on the preparedness? It does not make any sense. And, in the introduction it reads: "...which could intensity disease transmission". I do not think this makes any sense as well, do you mean "intensify"? The manuscript has to be reviewed and proof-read carefully to remove such errors and statements. Acronyms such as SEIR must be spelt in full in first appearance.

The manuscript has some inaccurate statements, which make me concerned about the quality of the literature review. For example, "Almost all studies pertained to earthquakes as a hazard with focus only on a single hospital as opposed to a complex network of hospitals 15". I appreciate that this reference is about 5 years old but I am concerned that the authors did failed to conduct a thorough and up to date literature. In fact, a substantial amount of research has been conducted about this: look at Aladhrai et al (2015) <https://doi.org/10.1017/dmp.2015.30>; and Achour and Miyajima (2020) <https://doi.org/10.1177/8755293020926180>. A thorough literature review is needed in order to provide more accurate statement.

The more serious issues for me are those related to the core of this research work.

The manuscript although is strong from theoretical and simulation point of views, it is weak from and practicality. Most of the work is based on assumptions, imagination and guessing which takes the realistic view of the actual problems of hospitals and healthcare systems. There are many more pressing problems that have not been well captured or at least recognised in this study (e.g. supply chains etc.). Where are these positioned in your research? WHO (2015) and researchers (e.g. Achour et al (2016) (<https://doi.org/10.1080/17477891.2016.1139539>) developed models for hospitals, how does your system link to these models? A healthcare system is a group of healthcare facilities / hospitals?

The combination of wildfire and COVID-19 is another issue that does not make much sense. This might be a possibility but currently it is just imagination. There are more realistic cases, which nature has proven such as that the mega-complex disaster of 11 March 2011 in Japan (earthquake, tsunami and nuclear leaks), I wonder why did you not look at this? How did you develop this scenario?

I am not convinced that wildfire is a 'natural disaster' as simply it can be an arson or accident so why would we blame it on nature while human has a major contribution to its occurrence.

The manuscript did not explain what data was used from your sample? How was this collected?

What is the analysis framework you followed? What is the definition of 'functionality' in this manuscript? How did you select the factors of this 'functionality'?

The manuscript even though is well written (from English perspective), it is vague and difficult to access. I am not sure how do you expect this to be read and understood by readers of this journal who often are people from different backgrounds and disciplines. You need to rethink the way it is written and ensure that it is easy and accessible to all people.

The extra documents you submitted are also too complicated. I do not think that readers will have the time or capacity to read and follow very complicated mathematical equations resolving some imaginative problems and using a number of assumptions. The manuscript needs to be re-developed thoroughly before considering it for publication.

Reviewer #2 (Remarks to the Author):

Comment to authors:

Please do accept my congratulations for connecting the dots and building a workable framework that looks into risks, disease propagation, patient behavior, and health care facility resilience for a well-rounded health system resilience analysis. Thank you for your wonderful manuscript. Please kindly consider my following suggestions.

Generic comments

I encourage authors to be more precise on the employed terminologies and to include appropriate definitions. An example is a definition of "quality of care", "standards of care" and employed indicators in the manuscript. Using the term of hospital capacity without indicating the metric explicitly. Please see my in-line comments on Supplementary Material.

Given the uncertainty of many input variables, overall I suggest including some stochastic analysis or at least sensitivity to some of the key assumptions to have a sense of the range of outputs.

Stating the key assumptions would shed light on interpreting the results and have to be included in the main body early on where needed. Please see my in-line comments. As I read through the document, I found some of my questions, however, some are quite buried and scattered. An example is that in methodology, wait times are aggregated for all patients types. However, one minute of wait for moderately-injured ER patients is very different from wait time to be admitted to ICU. If not addressed in terms of risks, I suggest to clarify this point in the main text and reflect on wait times that are also only at the point of entry of service and not through the patient care-path. The latter point is essential as when hospitals run out of inpatient beds, the wait time artificially increases in other units. The same goes for the Length of Stay for different treatment points for different patient types (routine ICU length of stay vs burnt patients vs COVID-19 patients).

The assumption of independence of hospital units (Inpatient beds and ICUs) is very strong and shown in the literature to be overlooking critical operational aspects of hospital capacity and capability management. I suggest to reflect on the gravity of this and reconsider the statement of lines 37 to 40 of the Supplementary material document. If hospital space is full including additional cots and every available space – there would be no place to transfer patients out from ICUs or admit patients from ER – hence the blockage will propagate upstream. Please refer to the body of literature (i.e. <https://doi.org/10.1080/24725579.2019.1584132>). I suggest the authors to discuss the suitability of this type of model for this particular study and use-case.

Being aware of the length of the manuscript, but equally important, is a discussion on the effects of Non-Pharmaceutical Interventions (NPIs) where the results are discussed. For example, how do we know that the lengthen peak of disease in the first compound scenario, wildfire 15 days before the expected peak, is a second wave as stated in line 192? We have failed to contain the diseases among evacuees in addition to the general public – the difference hence between first and second scenario (line 196) is mostly due to NPIs being in place during the peak as stated. I suggest considering to look into the "rates of change" for this part of the analysis as well.

The models and analysis are primarily for hospitals and not the other health care facilities such as primary health care that play an important role in lightening the burden from hospitals by accepting moderate patients. I refer the authors to South Korea health system response to COVID-19. Please clarify early on and distinguish between the potential use of the framework and the current work.

I recommend adding a statement on the take of this work on hospital capacity and capability with regards to routine, wildfire, and COVID-19 patients with key treatment factors for each patient type. This information is essential to interpret the results (i.e. Fig. 3). The reference is for only COVID-19 patients.

I understand that some information has been scattered throughout the reorganization of the manuscript. Please consider and overall check of definitions (i.e. functionality).

Finally, it would be interesting to see how the employed behavioral model differs from more commonly in use assumption of hospital choice as a function of distance and available #beds. Any comparison of results that you can share would be greatly of interest.

Main document:

Line 43: You can refer to Cyclone Amphan in Bangladesh and India.

Line 49: I suggest adding a few works of literature as a lot of works have been published on interdependent lifelines and impacts on services such as health (see for example [https://doi.org/10.1061/\(ASCE\)IS.1943-555X.0000465](https://doi.org/10.1061/(ASCE)IS.1943-555X.0000465))

Line 55: Burning and respiratory patients also require special care that can be beyond health care capacity and capability. The capability of the health system differs from capacity as a very clear case for critical COVID-19 patients in need of ICUs for example.

Line 103: Normalizing over population density draws a more clear picture than expectedly having a higher number of health care facilities in urban areas.

Line 107: as the study primarily focuses on hospitals it is beneficial to clarify the point as primary health care facilities, for example, have not been considered in the analysis for managing the surge demand.

Line 105: a clear definition of health care facilities that are included would be helpful. It seems that here authors mean hospitals as for example primary health care facilities are not expected to have ER but some are equipped with lab and imaging equipment.

Also, the care for burnt patients is very specialized to the point that some health system have specific hospitals dedicated to burnt patients. A brief discussion on the care needs of burnt patients and COVID-19 will be helpful to readers.

Line 114: do we know how much the capacity and capability were reduced?

Line 116: so H2 was not evacuated but not accessible at all? Addressed in line 177 but maybe works better here.

Line 138: do we know the increase and what hospital got the most portion of the surge?

Line 142: please include the % increase if before and after stats are available – the reference is to the website only.

Line 143: what is the definition of “health care system functionality” and “average wait time”? Do you refer wait time at the point of entry or through the patients’ care-path?

Line 144: “patient treatment time” is for specific services of emergency patients- regular ER patients versus emergency burnt patients? In other words, is this from point of entry to discharge on average for all patient types?

Line 146: by “inpatient demand” do you mean hospitalized patients? A stat on % of hospitalization need for wildfires would be useful to know. For example, only 20% of COVID-19 ER entries would need hospitalization, etc (please do not refer to the 20% number as it varies drastically).

Line 174: please explain quarantine in this context. Do you mean active cases? Or COVID-19 confirmed and home-care? or estimated overall COVID-19 cases?

Line 213: regular plus wildfire plus pandemic patients.

Line 229: what is the estimated maximum capacity given additional resources?

Starting line 218: analysis on crisis operational mode of hospitals: what is the assumed length of stay for critical patients with/without ventilators in ICUs? How it differs for wildfire patients? The length of stay for COVID-19 patients is almost 5 times of regular ICU stay which reduces the capacity over the study time horizon.

Line 286: how much increase the capacity (in terms of number) of shelters

Line 298: please give the accusation rate that is used.

Line 333: emphasizing the importance of assumptions used here otherwise hard to see how to achieve a temporary 300 bed hospital with 100+ ER and 25+ critical care units. Maybe use NYC examples on temporary tents here.

Line 342: 40 days after the outbreak for this set of scenarios would be quiet comparable with

wildfire 15 days before the peak as in the first series of outcomes?

On SEIR model scenarios: have you considered various hospitalization rates and critical care needs for different countries as well? It would be great to know the reasoning behind the aggregation of "estimated" active cases.

Supplementary material

Please include a brief literature review on hospital resiliency models that are of relevance in the supplementary material. Many works have used the fault/success trees as this model and a brief discussion on the benefit and suitability of this particular model would be of help.

S2: travel time of patients varies if patients are to enter ICUs directly from EMS or on foot to ER. So, the type of beds is important. As the model accounts for the quality of care, this will be nice to be reflected as also wait times translates to risk differently.

Line 58: treatment effect is under the general umbrella of Standards or Quality of care. You might have enough staff but short treatment time which is one indicator of Standard/Quality of care – however, in the formula you are tapping into "Doctor to patient" ratios which have other implications as well. Please consider alternative terminology for clarification.

Line 68: patient-driven model; recommend using something in lines of a patient-centric health-seeking behavior model.

Fig. S5: please indicate if the numbers on the exhibits are the "confirmed" cases or estimated/suspected cases.

Reviewer #3 (Remarks to the Author):

The paper reports a combined simulated impact of wildfire disaster and COVID-19 pandemic on healthcare system in a particular scenario. It seems to be built on a sound argument and based on rigorous analysis of data. The results demonstrate a degree of novelty, which will be of interest to researchers in the field who are keen to seek greater understanding of the combined impact of disasters on infrastructures. There are few issues which the authors may wish to address:

1. The efficacy of mitigation strategies of protection depends on individual and community adherence to the protective measures. This is related to behaviour (and socio-cultural) aspects of the measures, but should be considered in the modelling of the impact, perhaps on a scale of high, medium, and low adherence. It will be interesting to see how it will impact on the model.
2. It is unclear how the vulnerability of the community and healthcare system has been factored in the model. In terms of community, some members of community may be more vulnerable to COVID-19, and may be less mobile than others.
3. Presentation should be made clearer; particularly when the authors refer to sub-Figures, for example those in Figure 4. Due to complexity of the presentation of the figure and small font, they are difficult to follow, and appreciate the arguments/findings in the figures. They should be clarified to improve readership.
4. There are a lot of editing issues/errors in the manuscripts, for example: the word 'amble' (not sure whether this is correct?), 'Error! Reference source not found', missing 'of' between 'use' and 'ambulances'. The article should be carefully proof-read before submission.
5. Figure captions are very long. Can they be simplified, for readership?

Orchestrating Performance of Healthcare Networks Subjected to the Compound Events of Natural Disasters and Pandemic

Please find below our detailed response to each reviewers' comments. Unless otherwise specified, **the page numbers in the responses refer to the pages in the revised manuscript with highlighted changes. Changes made to address the comments appear in the manuscript in blue font.**

RESPONSE TO REVIEWER 1:

General Comments:

Thank you for this interesting submission. Glad to see more work is being conducted on the resilience of healthcare to disasters. The manuscript presents the findings of a set of simulations about the impact of fire and the pandemic COVID-19 on the resilience of healthcare systems. Although this an interesting point of view, there are some major and fundamental issues that have to be addressed before any recommendations for publication is given.

Authors Response: We sincerely appreciate all the time the reviewer spent in reading the manuscript and providing extensive and thoughtful comments. We modified the manuscript to address all comments and suggestions, as noted below, which has enabled us to improve the quality of the paper significantly.

Specific Comments:

1. The manuscript has many typos or maybe indications of lack of understanding of what the research is about. For example, in the abstract we find “Here, we investigate the combined impact of natural disasters and pandemic on the preparedness of a network of hospitals”. I am not sure how you can investigate the impact of disasters on the preparedness? It does not make any sense. And, in the introduction it reads: “...which could intensity disease transmission”. I do not think this makes any sense as well, do you mean “intensify”? The manuscript has to be reviewed and proof-read carefully to remove such errors and statements. Acronyms such as SEIR must be spelt in full in first appearance.

Authors Response: Thank you for your comments. We modified all the typos highlighted and we also proof-read the manuscript. We also made sure to define all acronyms before they appear in the paper. [please see page 1, line 17:18], [please see page 2, line 43], and [please see page 3, line 91].

2. The manuscript has some inaccurate statements, which make me concerned about the quality of the literature review. For example, “Almost all studies pertained to earthquakes as a hazard with focus only on a single hospital as opposed to a complex network of hospitals 15”. I appreciate that this reference is about 5 years old but I am concerned that the authors did failed to conduct a thorough and up to date literature. In fact, a substantial amount of research has been conducted about this: look at Aladhrai et al (2015) <https://doi.org/10.1017/dmp.2015.30>; and Achour and Miyajima (2020) <https://doi.org/10.1177/8755293020926180>. A thorough literature review is needed in order to provide more accurate statement.

Authors Response: We appreciate this comment as well and wish to thank the reviewer for pointing out these references. We would like to note that the 2020 reference by Achour and Miyajima was published in June 2020, which was after we had submitted our manuscript for review and this is the reason why it was not included in our initial submission. The 2015 paper by Aladhrai et al. is very interesting also and was likely missed because our focus was on natural disasters as noted in the first two sentences in that paragraph. Nevertheless, the two studies pointed out by the reviewer are interesting and we included them in our modified literature review. Although the two studies do not invalidate our statement on “almost all” studies on

hospitals conducted under natural disasters pertained to earthquakes and on single hospitals (i.e., studies on single hospitals under earthquakes overwhelms all other studies pertaining to healthcare facilities and natural disasters), to address the reviewer's comment, we added the two noted papers and removed the problematic statement to avoid any potential issues regarding the literature review. [please see page 2, line 74:76].

3. The more serious issues for me are those related to the core of this research work. The manuscript although is strong from theoretical and simulation point of views, it is weak from and practicality. Most of the work is based on assumptions, imagination and guessing which takes the realistic view of the actual problems of hospitals and healthcare systems. There are many more pressing problems that have not been well captured or at least recognised in this study (e.g. supply chains etc.). Where are these positioned in your research? WHO (2015) and researchers (e.g. Achour et al (2016) (<https://doi.org/10.1080/17477891.2016.1139539>) developed models for hospitals, how does your system link to these models? A healthcare system is a group of healthcare facilities / hospitals?

Authors Response: Thank you for raising the mentioned issues. We offer the response below to further explain our modeling approach and highlight that most of our assumptions were verified using published data and previous studies. Generally, our healthcare system model was based on a) experience gained from previous events, b) analytical and theoretical models introduced by other researchers as well as our own work, and c) logical relationships between the model components, which we then verified, in this study, with the data collected from a) California Department of Health and California Health and b) Human Services Open Data Portal (e.g. number of available staffed beds and the number of patients at each facility). The wildfire impact model inputs were collected from published data (e.g. Butte County Camp Fire Structure Status report). In addition, we constructed the disease transmission model based on the data published from different sources to obtain realistic estimate of the spread of the new coronavirus. We also considered uncertainties to estimate the number of pandemic-related patients. The numbers of regular patients were extracted from the data published by healthcare facilities during the Camp fire in 2018.

We also would like to thank the reviewer for his/her comment about what should be simulated in any comprehensive healthcare system model. We agree with the reviewer's point of view that many components have to be modeled or at least represented to reach an acceptable presentation of such a complicated system. We believe that adding more descriptions and discussions on the healthcare system model helped to clarify the necessary elements in this model. We also want to highlight that the utilized healthcare system model in this study was discussed in detail in a previous publications by the authors (*An integrated socio-technical approach for post-earthquake recovery of interdependent healthcare system & Full functionality and recovery assessment framework for a hospital subjected to a scenario earthquake event*).

The issue of supply chain is definitely important as pointed out by the reviewer and merits its own comprehensive study under the combined events. Future refinement of the utilized model in this study can for sure benefit from being integrated with a comprehensive supply-chain model. While our model did not include the supply chain network, supplies associated with Oxygen, Surgical, Rx, Fuel, Food, and other supplies are all included in a simplified way as shown in Fig. S1. We modified the manuscript to further highlight this issue and to link the utilized healthcare

system model with previous guidance and research work such as WHO (2015) and Achour et al (2016).

We agree with the reviewer's point of view that a healthcare system is a group of healthcare facilities, not just hospitals. However, we investigated hospitals in this study because they are the main healthcare provider, especially for wildfire and pandemic related patients. We modified the manuscript to list the reason for selecting hospitals in this study and indicated that in some cases other healthcare facilities should be included in the network analysis. [please see page 13, line 442:451], [please see page 15, line 504], and [please see page 15, line 520:527].

4. The combination of wildfire and COVID-19 is another issue that does not make much sense. This might be a possibility but currently it is just imagination. There are more realistic cases, which nature has proven such as that the mega-complex disaster of 11 March 2011 in Japan (earthquake, tsunami and nuclear leaks), I wonder why did you not look at this? How did you develop this scenario?

Authors Response: The reviewer raises an important point that merits further clarifications. Undoubtedly, the most pressing hazards to include for a specific region should be those in which high-intensity events exist with small return periods. In the case of pandemics, the return period worldwide is about 100 years (Fan et al. 2017 <http://dx.doi.org/10.2471/BLT.17.199588>), which is pretty low (meaning major pandemics occur relatively very often). Therefore, combining pandemics with a natural hazard is a very plausible scenario, particularly if that hazard has a low return period as well. Wildfires have been on the rise in recent years and their intensity and frequency have increase (Schoennagel et al. 2017 <https://doi.org/10.1073/pnas.1617464114>). Recent studies on wildfire occurrence in a specific region have shown their return period to be very small. For the area of interest (i.e. Northern California), the return period is approximately 35-200 years (please see Chapter 3 on Fire Regimes in the following publication https://www.firescience.gov/JFSP_fire_history.cfm), which is much lower than significant earthquakes. Accordingly, the likelihood of pandemics and wildfires is substantially high. This is demonstrated by the recent and current exposure of many regions in the U.S. to both events (pandemics and wildfires) simultaneously (e.g., Colorado and California have been dealing with this issue for the past few months). Accordingly, the two disasters are now at the forefront of the research issues to be addressed and one of the main topics addressed by federally funded agencies such as the CDC and the US Department of Interior. We agree that assessing the impact of earthquakes and tsunami on communities and the potential leak from a nuclear facility is also a very interesting problem to evaluate. Such an event would be considered a low probability high consequence and could be of interest to policymakers and healthcare officials. We modified the manuscript to highlight the importance of investigating both disasters and we also discussed the development of the two scenarios. [please see page 2, line 34:35] and [please see page 3, line 74:76]

5. I am not convinced that wildfire is a 'natural disaster' as simply it can be an arson or accident so why would we blame it on nature while human has a major contribution to its occurrence.

Authors Response: Thank you for raising this great point as well. We would like to clarify that we agree that the initiation of the fires in the wildland are mostly man-made. The spread of fire, however, is a natural phenomenon. This was recently (spring 2018) acknowledge by the Wildfire Disaster Funding Act (H.R. 2862) passed by U.S. Congress where wildfires are now classified as

a natural hazard just like hurricanes, floods, and earthquakes. Similarly, many other organizations such as World Health Organizations (WHO)- (https://www.who.int/environmental_health_emergencies/natural_events/en/) classify wildfire as a natural disaster as quoted “*Natural disasters include earthquakes, tsunamis, volcanic eruptions, landslides, hurricanes, floods, wildfires, heat waves and droughts.*” We modified the manuscript to clarify this issue. [please see page 2, line 57:61]

6. The manuscript did not explain what data was used from your sample? How was this collected? What is the analysis framework you followed? What is the definition of ‘functionality’ in this manuscript? How did you select the factors of this ‘functionality’?

Authors Response: Thank you for this comment. We modified the manuscript to address all the comments raised where we a) provided a more clear description about the data we utilized in our analysis, b) added more discussion about the data sources used, c) clearly linked the methodology section with the utilized model listed in the supplementary material, d) provided a more detailed definition of the functionality term and e) clarified the selection of the functionality factors and linked it to the previous guidelines and research studies that we used to build this framework. [please see page 13, line 422:451], [please see page 17, line 588:609], and [please see Table S1, Table S2, and Table S3 in the SI document].

7. The manuscript even though is well written (from English perspective), it is vague and difficult to access. I am not sure how do you expect this to be read and understood by readers of this journal who often are people from different backgrounds and disciplines. You need to rethink the way it is written and ensure that it is easy and accessible to all people.

Authors Response: Thank you for raising this issue. We sincerely appreciate this reflection. We have revised the manuscript to ensure that the language and style used are appropriate for readers with diverse backgrounds.

8. The extra documents you submitted are also too complicated. I do not think that readers will have the time or capacity to read and follow very complicated mathematical equations resolving some imaginative problems and using a number of assumptions. The manuscript needs to be re-developed thoroughly before considering it for publication.

Authors Response: Thank you for this comment. We modified the Supplementary Information document and provided simplified descriptions of the utilized models so that the paper is accessible to general readers and also clarified the descriptions of the mathematical equations so that the reproduction of this study by any researcher is possible.

RESPONSE TO REVIEWER 2:

Generic Comments:

Please do accept my congratulations for connecting the dots and building a workable framework that looks into risks, disease propagation, patient behavior, and health care facility resilience for a well-rounded health system resilience analysis. Thank you for your wonderful manuscript. Please kindly consider my following suggestions.

Authors Response: We sincerely appreciate the very kind and encouraging words about the manuscript. We also thank the reviewer for all the time she/he spent reading the manuscript and providing extensive and thoughtful comments. We acknowledge that addressing the comments, as noted below, has enabled us to significantly improve the quality of the manuscript.

Specific Comments:

1. I encourage authors to be more precise on the employed terminologies and to include appropriate definitions. An example is a definition of “quality of care”, “standards of care” and employed indicators in the manuscript. Using the term of hospital capacity without indicating the metric explicitly. Please see my in-line comments on Supplementary Material.

Authors Response: Thank you for your comment. We modified the manuscript to clearly defined the terminologies we utilized as well as the metric we used to measure them. [please see page 13, line 422:451] and [please see page 3, line 78:79].

2. Given the uncertainty of many input variables, overall I suggest including some stochastic analysis or at least sensitivity to some of the key assumptions to have a sense of the range of outputs.

Authors Response: Thank you for this comment. We agree that an uncertainty or sensitivity analysis could provide valuable insight on the study. We followed up on your suggestion and utilized statistical distributions for different hospitalization rates (infections requiring hospitalization and hospitalizations admitted to ICU) for various age groups (from <10y to >90y), which were estimated based on the data published by ECDC for different European countries during the period from the pandemic outbreak to June 30th, 2020. We summarized and discussed these distributions along with other statistical distribution for the patients requiring ventilation in the Supplementary Information (SI). We then used uncertainty propagation analysis to estimate a range of the expected number of pandemic-related patients and the expected demand on healthcare facilities. Knowing that the numbers of regular and wildfire-related patients are extracted from the data published by healthcare facilities during the Camp fire in 2018. Finally, we used the Monte-Carlo simulation with 100,000 trials to estimate the confidence intervals. **Fig. 2** was then modified to draw the 2.5 and 97.5 percentiles of the patient distribution per facility. We also added a new figure in the SI to show the uncertainty in the estimated number of COVID-19 related patients. [please see page 17, line 588:594], [please see Fig. 2], [please see Fig. S6 in the SI document], and [please see Table S3 in the SI document].

3. Stating the key assumptions would shed light on interpreting the results and have to be included in the main body early on where needed. Please see my in-line comments. As I

read through the document, I found some of my questions, however, some are quite buried and scattered. An example is that in methodology, wait times are aggregated for all patients types. However, one minute of wait for moderately-injured ER patients is very different from wait time to be admitted to ICU. If not addressed in terms of risks, I suggest to clarify this point in the main text and reflect on wait times that are also only at the point of entry of service and not through the patient care-path. The latter point is essential as when hospitals run out of inpatient beds, the wait time artificially increases in other units. The same goes for the Length of Stay for different treatment points for different patient types (routine ICU length for stay vs burnt patients vs COVID-19 patients).

Authors Response: Thank you for this comment and the added clarifications. We modified the manuscript to list each models' parameters and the methods used to calculate them as early as possible in the methodology section. We would like also to thank you for raising the issue of the waiting time calculations. We had stated in the SI document that the waiting time is only for the patients before seeing a medical care provider but not for the internal medical services, which we acknowledge the importance of such. While we did not explicitly account for the risk associate with the wait time, we used internal treatment time as a metric to reflect on the quality of the internal services. We then combined the two indicators to measure the overall quality of the provided service. We also would like to highlight that we considered hospitals to follow a dynamic triage that will ensure that the patients who are in critical conditions will receive the required medical service sooner. For the length of stay, we used an average length of stay for each patient category considered in our study. These numbers were collected from published data from various sources as noted in the paper. We then modified the length of stay values that we collected to account for the delay that might result from transferring patients to another hospital unit (e.g., patient needs to be admitted to the hospital but has to wait in the ER until an inpatient bed becomes available). We modified the manuscript to clarify the mentioned points and illustrate the modeling of the patient's length of stay and highlighted the factors that impact its calculations. [please see page 13, line 437:443] and [please see page 14, line 599:607].

4. The assumption of independence of hospital units (Inpatient beds and ICUs) is very strong and shown in the literature to be overlooking critical operational aspects of hospital capacity and capability management. I suggest to reflect on the gravity of this and reconsider the statement of lines 37 to 40 of the Supplementary material document. If hospital space is full including additional cots and every available space – there would be no place to transfer patients out from ICUs or admit patients from ER – hence the blockage will propagate upstream. Please refer to the body of literature (i.e. <https://doi.org/10.1080/24725579.2019.1584132>). I suggest the authors to discuss the suitability of this type of model for this particular study and use-case.

Authors Response: Yes, we totally agree with the reviewer on this important point. We acknowledge the significance of the study presented by TariVerdi et al. as one of the detailed approaches to model hospital units and we used it as a reference when we developed our model. We would like to also note that we considered hospital units as independent in terms of resource management (staff, space, and supplies). We, however, considered their interaction in relation to transfer resources and/or patients including decisions based on the situation in other units. For example, if all units are full except the ER, then the ER will continue to accept patients, but the patients will not be admitted to the hospital. In this case, patients can a) receive treatment in the

ER or wait if their case is not critical, b) transfer to another hospital, or c) be sent home and considered as untreated patients. We modified the manuscript to clarify our work further and acknowledge the limitations of the model in terms of the level of interaction being captured between the different units. [please see page 13, line 443:451] and [please see page 2, line 57:68 in SI].

5. Being aware of the length of the manuscript, but equally important, is a discussion on the effects of Non-Pharmaceutical Interventions (NPIs) where the results are discussed. For example, how do we know that the lengthen peak of disease in the first compound scenario, wildfire 15 days before the expected peak, is a second wave as stated in line 192? We have failed to contain the diseases among evacuees in addition to the general public – the difference hence between first and second scenario (line 196) is mostly due to NPIs being in place during the peak as stated. I suggest considering to look into the “rates of change” for this part of the analysis as well.

Authors Response: Thank you for this comment. We again agree with the reviewer’s point of view and would like to note that the results of the modified SEIR model that we used show an increase of the Non-Pharmaceutical Interventions (in the form of masks, social distancing, and business and schools lockdown) over time after the initial outbreak of the COVID-19 as a result of the strong measures applied to control the disease spread. We also discuss the reproduction number as an indicator of the rate of change in the number of infected cases when we discuss the compounded impact of wildfire and pandemic. We modified the manuscript to add more discussions related to the Non-Pharmaceutical Interventions and clarify our consideration of the rate of change in the number of infected cases. [please see page 6, line 215:239].

6. The models and analysis are primarily for hospitals and not the other health care facilities such as primary health care that play an important role in lightening the burden from hospitals by accepting moderate patients. I refer the authors to South Korea health system response to COVID-19. Please clarify early on and distinguish between the potential use of the framework and the current work.

Authors Response: Thank you for this comment and we agree with your reflection. We included non-acute healthcare facilities but not all other healthcare facilities. We modified the manuscript to clearly state that we only focused on hospitals and their associated non-acute healthcare facilities while acknowledging the role of other healthcare facilities for treating moderate cases. [please see page 13, line 422:431].

7. I recommend adding a statement on the take of this work on hospital capacity and capability with regards to routine, wildfire, and COVID-19 patients with key treatment factors for each patient type. This information is essential to interpret the results (i.e. Fig. 3). The reference is for only COVID-19 patients.

Authors Response: Thank you for this comment. We modified the manuscript to show the difference between the hospital's capacity and capability to treat different patients. We also would like to highlight that the data shown in **Fig. 3** (and also other figures in the main document) are for the total number of patients and not for COVID-19 related patients only. The patient distribution for the wildfire and regular patients are shown in **Fig. S4** in the SI and the COVID-19 related patients and regular patients are shown in **Fig. S6** in the SI. [please see page 5, line 185:190].

8. I understand that some information has been scattered throughout the reorganization of the manuscript. Please consider and overall check of definitions (i.e. functionality).

Authors Response: Thank you for this comment. We modified the manuscript to make sure all terms are clearly defined. [please see page 13, line 422:431].

9. Finally, it would be interesting to see how the employed behavioral model differs from more commonly in use assumption of hospital choice as a function of distance and available #beds. Any comparison of results that you can share would be greatly of interest.

Authors Response: Thank you for raising this issue. To address your comment, we introduced a comparison between the proposed patient-driven model in this study and a model presented by Jia et al. (2019) which was developed based on 2,376,743 inpatient discharge records from 22 acute long-term care hospitals and 199 general medical and surgical hospitals. The comparison is shown in the SI document. We also would like to highlight that we verified the introduced patient-driven model with the data collected from Butte County during the Camp Fire incident. We also included a literature review about the hospital choice models in general and how it relates to the patient-driven model we utilized in this study. [please see page 3, line 103:112 in the SI], [please see page 9, line 205:213 in the SI], and [please see Fig. S8 in the SI].

In-line Comments:

Main document:

1. Line 43: You can refer to Cyclone Amphan in Bangladesh and India.

Authors Response: Thank you for this comment. We referred to this incident as an example of coupling natural disasters and disease.

2. Line 49: I suggest adding a few works of literature as a lot of works have been published on interdependent lifelines and impacts on services such as health (see for example [https://doi.org/10.1061/\(ASCE\)IS.1943-555X.0000465](https://doi.org/10.1061/(ASCE)IS.1943-555X.0000465))

Authors Response: Thank you for this comment. We updated our literature review regarding the interdependency, and we added the work done by TariVerdi et al., Jacques et al., and Cimellaro as examples. [please see page 2, line 49].

3. Line 55: Burning and respiratory patients also require special care that can be beyond health care capacity and capability. The capability of the health system differs from capacity as a very clear case for critical COVID-19 patients in need of ICUs for example.

Authors Response: Thank you and we agree with your comment. We modified the manuscript to include burning patients after the wildfire. With that said, out of the total burn-related patients after Camp Fire (19 civilians and 5 firefighters), only two patients were treated in Butte County. Based on the data published by American Burn Association (ABA) no certified burn centers exist in Butte County, which was the reason why those patients received treatment in other medical centers outside the investigated county such as UC Davis Medical Center in Sacramento and St. Francis Memorial Hospital. Even though we totally agree with the reviewer comment, in this case, none of the investigated hospitals housed burn treatment units and the number of burn-related patients was not significant. Therefore, we modified the manuscript to reflect on burn injuries and the importance of modeling burn centers discussions. We also distinguish between

the capacity and capability in the manuscript as noted by the reviewer. [please see page 2, line 52:55] and [please see page 15, line 506:510].

4. Line 103: Normalizing over population density draws a more clear picture than expectedly having a higher number of health care facilities in urban areas.

Authors Response: We appreciate your comment. We modified **Fig. 1** to include the normalized hospital number over the population. [please see page 3, line 111:113] and [please see Fig. 1].

5. Line 107: as the study primarily focuses on hospitals it is beneficial to clarify the point as primary health care facilities, for example, have not been considered in the analysis for managing the surge demand.

Authors Response: Thank you for this comment. We modified the manuscript to clarify this issue as per our response to Specific Comment #6. [please see page 13, line 422:431].

6. Line 105: a clear definition of health care facilities that are included would be helpful. It seems that here authors mean hospitals as for example primary health care facilities are not expected to have ER but some are equipped with lab and imaging equipment. Also, the care for burnt patients is very specialized to the point that some health system have specific hospitals dedicated to burnt patients. A brief discussion on the care needs of burnt patients and COVID-19 will be helpful to readers.

Authors Response: Thank you for the comment. We modified the manuscript as suggested. [please see page 3, line 114:122].

7. Line 114: do we know how much the capacity and capability were reduced?

Authors Response: Yes, this information was partially discussed in the results section in the original submission. Overall, the reduction in capacity was approximately 17% [please see page 5, line 168:170, and Page 15, line 494:496]. We modified the paper to also reflect on the reduction in capability [please see page 4, line 126:131, and page 5, line 161:169].

8. Line 116: so H2 was not evacuated but not accessible at all? Addressed in line 177 but maybe works better here.

Authors Response: Thank you for your comment. Feather River Hospital was evacuated after the wildfire and remained inaccessible during the wildfire. After the wildfire, damage assessment for the hospital building indicated severe damage to many of the hospital buildings on campus, which prevented the hospital reopening [please see the following publication: Wildfire impacts on schools and hospitals following the 2018 California Camp Fire]. We modified the manuscript to clarify this issue. [please see page 4, line 126:129].

9. Line 138: do we know the increase and what hospital got the most portion of the surge?

Authors Response: Yes, the distribution of the increase in patient demand was discussed in the original manuscript and is also shown in **Fig. S4** in the SI. The surge in demand was 14%, 13%, and 15% at the Enloe, the Orchard Hospital, and the Oroville Medical Center, respectively compared with their demand before the wildfire. [please see page 4 line 165:168].

10. Line 142: please include the % increase if before and after stats are available – the reference is to the website only.

Authors Response: Thank you. The % values have been added to the manuscript and the reference has been modified. [please see page 5 line 158:161].

11. Line 143: what is the definition of “health care system functionality” and “average wait time”? Do you refer wait time at the point of entry or through the patients’ care-path?

Authors Response: We appreciate your comment. We addressed this question in our response to Specific Comments # 1, 6, and 8.

12. Line 144: “patient treatment time” is for specific services of emergency patients- regular ER patients versus emergency burnt patients? In other words, is this from point of entry to discharge on average for all patient types?

Authors Response: Thank you for raising this issue. As we mentioned in our response to the reviewer's Specific Comment #3, the waiting time we considered in this study was the initial waiting time and we used treatment time as another metric to measure the quality of the service for patients who are admitted to the hospital.

13. Line 146: by “inpatient demand” do you mean hospitalized patients? A stat on % of hospitalization need for wildfires would be useful to know. For example, only 20% of COVID-19 ER entries would need hospitalization, etc (please do not refer to the 20% number as it varies drastically).

Authors Response: Thank you for this comment. We meant the number of patients admitted to the hospital and treated in the inpatient unit. we modified the manuscript to show the percentage increase in patients after the wildfire, which we categorized here as burning and respiratory patients. [please see page 5, line 154:158].

14. Line 174: please explain quarantine in this context. Do you mean active cases? Or COVID-19 confirmed and home-care? or estimated overall COVID-19 cases?

Authors Response: Thank you for raising this issue. Quarantine represents all COVID-19 patients who are infected and confirmed (tested positive). We defined the different stages of the disease transmission in the methodology section and we modified the manuscript to clearly define the quarantine. [please see page 6, line 196:198].

15. Line 213: regular plus wildfire plus pandemic patients.

Authors Response: Thank you for the note. We modified the text accordingly. [please see page 6, line 213:215] and [please see page 7, line 246:250].

16. Line 229: what is the estimated maximum capacity given additional resources?

Authors Response: Thank you for this comment. The increased capacity for ER departments is 600, 30, and 200 at H1, H3, and H4, respectively, we modified the manuscript to highlight the increase in capacity after considering all the resources. [please see page 8, line 272:274].

17. Starting line 218: analysis on crisis operational mode of hospitals: what is the assumed length of stay for critical patients with/without ventilators in ICUs? How it differs for wildfire patients? The length of stay for COVID-19 patients is almost 5 times of regular ICU stay which reduces the capacity over the study time horizon.

Authors Response: Thank you for this comment. We did consider different lengths of stay for different patient categories. These numbers were collected from the published data from various sources as indicated in the citation. We modified the manuscript to not only include the citation but also the actual numbers. [please see page 18, line 599:607].

18. Line 286: how much increase the capacity (in terms of number) of shelters

Authors Response: Thank you for this comment. We suggest an increase in the number of shelters until a safe social distance of 6-ft between all individuals inside the shelter, including residents and staff working. We modified the manuscript to indicate this point. [please see page 10, line 325:328].

19. Line 298: please give the accusation rate that is used.

Authors Response: The rate is 19%. We modified the paper accordingly. [please see page 10, line 337:339].

20. Line 333: emphasizing the importance of assumptions used here otherwise hard to see how to achieve a temporary 300 bed hospital with 100+ ER and 25+ critical care units. Maybe use NYC examples on temporary tents here.

Authors Response: We appreciate your comment. We would like to clarify that the ratio between the bed types in **Fig. 4(f)**, is not very typical for a conventional hospital. Your suggestion on using NYC as an example is great and we modified the paper accordingly. [please see page 11, line 375:378].

21. Line 342: 40 days after the outbreak for this set of scenarios would be quite comparable with wildfire 15 days before the peak as in the first series of outcomes?

Authors Response: Thank you for this comment. Yes, for scenario *S6* 40 days after the outbreak is the same as 15 days before the peak. To make that clear we modified the manuscript to show that the scenario we used in the discussion section is one of the scenarios we showed in the results sections. To make this part clear we also added the peak time for each scenario in **Fig. S6**. [please see page 9, line 310] and [please see Fig. S7 in SI document].

22. On SEIR model scenarios: have you considered various hospitalization rates and critical care needs for different countries as well? It would be great to know the reasoning behind the aggregation of "estimated" active cases.

Authors Response: In this study, we modeled different disease spread scenarios and as requested by the reviewer, we also considered the uncertainty associated with the hospitalization rates. We aggregated all the different active cases to compare it with the data we collected for the confirmed cases that were published by Dong et al. We subcategorized the active cases to differentiate between the infective cases who are not yet quarantined and the quarantined cases and to estimate the number of patients in each category based on their medical needs. Different categories were included to subcategorized the active cases including a) infective (infected but not confirmed and can spread the disease as they are not quarantined yet), b) quarantined (infected and confirmed but with asymptomatic or with mild symptoms and do not need medical care), c) hospital admitted (either inpatient, ICU, or mechanical ventilator). We modified the manuscript to clearly show the reason behind aggregating and subcategorizing the active cases. [please see page 17, line 588:594] and [please see page 15, line 520:527].

Supplementary material:

23. Please include a brief literature review on hospital resiliency models that are of relevance in the supplementary material. Many works have used the fault/success trees as this model and a brief discussion on the benefit and suitability of this particular model would be of help.

Authors Response: Thank you for this comment. We modified the SI to cover more a literature review about the resilience of the healthcare system and the use of a fault tree in risk analysis. We also briefly discussed the basics of the introduced model in the manuscript. [please see page 1, line 13:28 in SI document].

24. S2: travel time of patients varies if patients are to enter ICUs directly from EMS or on foot to ER. So, the type of beds is important. As the model accounts for the quality of care, this will be nice to be reflected as also wait times translates to risk differently.

Authors Response: Thank you for this comment. We modified the manuscript to highlight the role played by this factor in calculating the waiting time. [please see page 3, line 82:85 in SI].

25. Line 58: treatment effect is under the general umbrella of Standards or Quality of care. You might have enough staff but short treatment time which is one indicator of Standard/Quality of care – however, in the formula you are tapping into “Doctor to patient” ratios which have other implications as well. Please consider alternative terminology for clarification.

Authors Response: Thank you for raising this point which is well acknowledged. We modified the paper (and re-ran our analysis to included space-to-patient ratio). [please see page 2, line 69:73 in SI document] and [please see page 3, line 93:100 in SI].

26. Line 68: patient-driven model; recommend using something in lines of a patient-centric health-seeking behavior model.

Authors Response: Thank you for this comment. We briefly discussed the relation between the patient-driven model and the patient-centric health-seeking model from the literature and we enhanced the literature review about these models. [please see page 3, line 103:110 in the SI document].

27. Fig. S5: please indicate if the numbers on the exhibits are the “confirmed” cases or estimated/suspected cases.

Authors Response: We appreciate your comment. We modified the figure description to highlight that the confirmed numbers shown in this figure represent the cases tested positive. [please see page 6, line 180:182 in SI document].

RESPONSE TO REVIEWER 3:

General Comments:

The paper reports a combined simulated impact of wildfire disaster and COVID-19 pandemic on healthcare system in a particular scenario. It seems to be built on a sound argument and based on rigorous analysis of data. The results demonstrate a degree of novelty, which will be of interest to researchers in the field who are keen to seek greater understanding of the combined impact of disasters on infrastructures. There are few issues which the authors may wish to address:

Authors Response: We appreciate all the thoughtful comments and the encouraging words that the reviewer provided. We acknowledge that addressing these comments along with comments from other reviewers has enabled us to improve the quality of the paper significantly.

Specific Comments:

1. The efficacy of mitigation strategies of protection depends on individual and community adherence to the protective measures. This is related to behaviour (and socio-cultural) aspects of the measures, but should be considered in the modelling of the impact, perhaps on a scale of high, medium, and low adherence. It will be interesting to see how it will impact on the model.

Authors Response: Thank you for raising this excellent point. Definitely, the socio-cultural and economic fabrics of communities will have large impact on the level of adherence to the various measures. We also have done a recent study (under review) on the impact of mitigation strategies and different levels of exposure and protective measure on the spread of the COVID-19 disease in the US, in which we discussed this issue from the healthcare system perspective. (*Impact of COVID-19 Second Wave on Healthcare Networks in the United States* <https://www.medrxiv.org/content/10.1101/2020.07.11.20151217v1>). To address this comment and to relate to our new study, we modified the manuscript to highlight the impact of community adherence to the protective measures on disease spread. [please see page 13, line 409:411].

2. It is unclear how the vulnerability of the community and healthcare system has been factored in the model. In terms of community, some members of community may be more vulnerable to COVID-19, and may be less mobile than others.

Authors Response: Thank you for your comment. The vulnerability of different community members was accounted for in the disease transmission model by simulating individuals in each age group separately (10 age groups were considered up to +90 years in age) and assigning hospitalization rates to each group.

3. Presentation should be made clearer; particularly when the authors refer to sub-Figures, for example those in Figure 4. Due to complexity of the presentation of the figure and small font, they are difficult to follow, and appreciate the arguments/findings in the figures. They should be clarified to improve readership.

Authors Response: Thank you for this comment. We modified the figures and we used a readable font for all of them to make them all clear.

4. There are a lot of editing issues/errors in the manuscripts, for example: the word 'amble' (not sure whether this is correct?), 'Error! Reference source not found', missing 'of' between 'use' and 'ambulances'. The article should be carefully proof-read before submission.

Authors Response: Thank you for this comment. We conducted a detailed proofread of the manuscript and the SI and we corrected all editorial issues.

5. Figure captions are very long. Can they be simplified, for readership?

Authors Response: Thank you for this comment. We simplified and shortened the figure captions as much as we can while making sure the caption sufficiently describes the content of the figure.

REVIEWERS' COMMENTS

Reviewer #1 (Remarks to the Author):

I am happy with the revisions that Authors made.

Reviewer #2 (Remarks to the Author):

I thank the authors for reflecting on all my extensive list of suggestions. I want to acknowledge the substantial time you spent and the effort to include additional analysis layers and further clarifications to both the manuscript and SI. I am looking forward to your upcoming works on the topic.

Regards,
Mersedeh Tariverdi, Ph.D.